# Neural dynamics of semantic control underlying generative storytelling
Clara Rastelli [1,2,3,4] ✉, Antonino Greco[2,3,4], Chiara Finocchiaro[1], Gabriele Penazzi [1],
Christoph Braun[2,3,4] & Nicola De Pisapia [1] ✉

Storytelling has been pivotal for the transmission of knowledge across human history, yet the role of semantic control and its associated neural dynamics has been poorly investigated. Here, human participants generated stories that were either appropriate (ordinary), novel (random), or balanced (creative), while recording functional magnetic resonance imaging (fMRI). Deep language models confirmed participants adherence to task instructions. At the neural level, linguistic and visual areas exhibited neural synchrony across participants regardless of the semantic control level, with parietal and frontal regions being more synchronized during random ideation. Importantly, creative stories were differentiated by a multivariate pattern of neural activity in frontal and fronto-temporo-parietal cortices compared to ordinary and random stories. Crucially, similar brain regions were also encoding the features that distinguished the stories. Moreover, we found specific spatial frequency patterns underlying the modulation of semantic control during story generation, while functional coupling in default, salience, and control networks differentiated creative stories with their controls. Remarkably, the temporal irreversibility between visual and high-level areas was higher during creative ideation, suggesting the enhanced hierarchical structure of causal interactions as a neural signature of creative storytelling. Together, our findings highlight the neural mechanisms underlying the regulation of semantic exploration during narrative ideation.

The generation of narratives has been pivotal for the transmission of knowledge, values, and cultural norms across human history[1,2]. It is through stories that we make sense of the world around us shaping our understanding of reality[3–6]. This intrinsic connection between storytelling and the human experience underscores not just the importance of narratives in preserving cultural heritage[7–9], but also in the construction and expression of the self[10–12].

Generating stories can be regarded as a context-dependent decision-making process that operates in a semantic space[13] by orchestrating the selection of linguistic units across multiple levels of abstraction[14], from single words to sentence-level attributes such as a coherent causal structure among the narrated events[15]. Crucial for this process is the ability to exert cognitive control on the semantic representations stored in memory[16–19], a phenomenon that has been named semantic control in the literature[20,21]. Recent evidence has shown how semantic control is implemented across a large-scale distributed pattern of brain networks activity[22] involving left fronto-temporal areas such as inferior frontal gyrus (IFG), posterior middle temporal gyrus (pMTG), and dorsomedial prefrontal cortex (dmPFC)[16,17,23,24].

This semantic control network is also partially distinct from brain areas encoding semantic representations[17,21], although they are frequently interacting during both the encoding and retrieval of semantic information[16,19]. The semantic representation system, centered in the anterior temporal lobes (ATLs) and linked to the default mode network (DMN), enables automatic retrieval and abstraction, forming stable, context-independent representations essential for storytelling[16–19,25,26]. Moreover, a recent meta-analysis confirmed the involvement of these areas in semantic control and found no implication of parietal areas[20], although the inclusion criteria incorporated a large cohort of studies including visual and auditory paradigms. In the context of linguistic tasks, semantic control was found to be implemented across similar fronto-temporal as well as parietal areas, especially in tasks requiring the generation of divergent associations to linguistic cues[27–30].

One of the most important questions related to storytelling is what makes a story interesting or noteworthy. To put it simply, how can we define a story "creative"? Linguistic creativity, defined as the ability to generate novel and appropriate ideas, has been reframed through computational and semantic frameworks as optimal functioning within a semantic space,

[1]Department of Psychology and Cognitive Science, University of Trento, Rovereto, Italy. [2]MEG Center, University of Tübingen, Tübingen, Germany. [3]Department of Neural Dynamics and Magnetoencephalography, Hertie Institute for Clinical Brain Research, University of Tübingen, Tübingen, Germany. [4]Centre for Integrative Neuroscience, University of Tübingen, Tübingen, Germany. ✉e-mail: clara.rastelli@uni-tuebingen.de; nicola.depisapia@unitn.it

balancing exploration and exploitation[31–40]. The neuroscience of creative cognition suggests that multiple large-scale networks support the generation and refinement of ideas. Generally, studies[41,42] have found that creative language production involves a balance between spontaneous processes, mostly linked to the DMN, and controlled processes, associated with the Executive Control Network (ECN). Despite the definition of creativity requires both the novelty and the appropriateness components, nearly all neuroscientific studies of linguistic creative ideation have focused only in comparing creative versus ordinary generations, thus investigating only the novelty component. Few studies at the behavioral level have also compared creative solutions with random ones, effectively studying the appropriateness component and the underlying semantic control processes[31,43].

Moreover, despite the extensive literature investigating the neural mechanisms of semantic control in generative language tasks, most of the studies have implemented experimental paradigms involving basic linguistic outputs such as single words or simple sentences[44]. Similarly in the field of linguistic creativity[41,45–47], most of the previous investigations adopted experimental paradigms with simple word associations to study the exploration of semantic space for finding novel and appropriate (i.e., creative) solutions to a given task[29,31,48–50]. This lack of ecological validity may be ascribed to the difficulty in the implementation and data analysis of complex linguistic objects such as narratives[51]. Recently, the advent of deep neural network models[52] applied to natural language processing has revolutionized our ability to analyze and quantify semantic information in textual data.

These models, typically trained on large-scale linguistic corpora, provide high-dimensional vector representations, or embeddings, that capture semantic relationships across multiple levels, from single words to entire sentences[53,54]. By incorporating contextual information, they can represent words and phrases as points in a continuous vector space, illuminating subtle distinctions that are especially relevant for understanding and modeling narratives.

Due to this difficulty in quantifying large textual data and contextualized semantic information before deep language models, the neuroscientific study of generative storytelling has been scarce. For example, one of the basic paradigms involving the generation of stories in the literature requires participants to encode stories and then report them[14,55–57]. Although this can be technically described as generative storytelling, it is quite restrictive due to the severe external constrains given to the participants by requiring them to faithfully report a previously presented story. Nevertheless, there were a few exceptions of studies involving unconstrained generation of stories in their paradigms[40,58–61]. Howard-jones and colleagues[58] found increased prefrontal and cingulate cortices activity by contrasting neural activations underlying creative and uncreative stories. Shah et al.[59] found a large parieto-fronto-temporal network underlying the ideation of creative stories, while Erhard et al.[60] reported the left frontal cortex and occipital cortex as being mostly activated during ideation in expert writers. Similarly, Liu et al.[61] reported a parieto-fronto-temporal network underlying the generation and writing of poems. Additionally, Fan

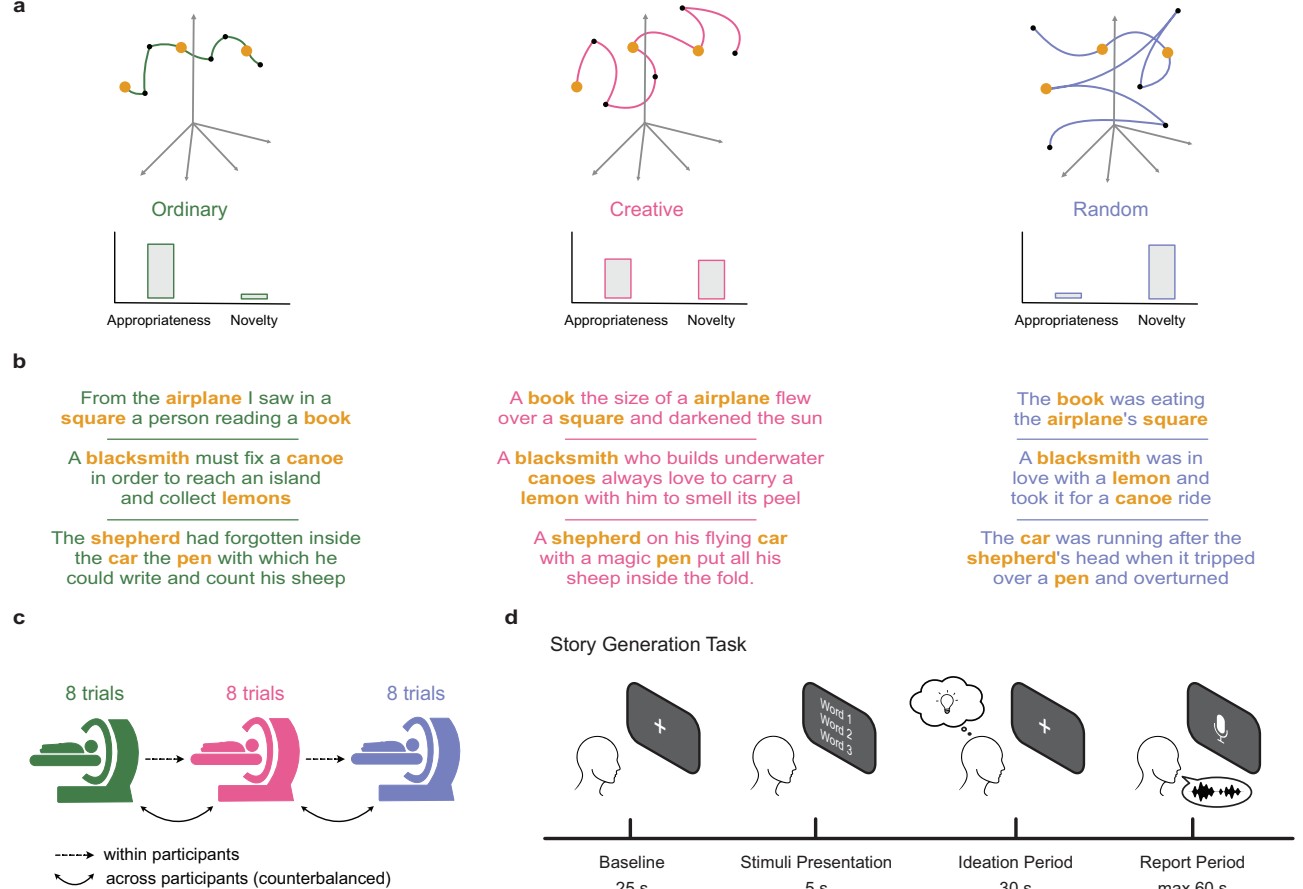

**Fig. 1 | Experimental design of our story generation task. a** Graphical representation of the underlying generative process in the semantic space among the three different conditions, namely the Ordinary (green), Creative (pink), and Random (blue) conditions. Participants were instructed to modulate the appropriateness and novelty of the story, given three target words. Orange dots represent the target words, while black dots stand for the other words composing the story. **b** Examples of generated stories sampled from different participants given the same triplet of target words (bolded orange). **c** Graphical depiction of the experimental paradigm. Each condition was composed by 8 trials (stories) in a block fashion. The order of conditions, as well as the assignment of the target words to the conditions, was counterbalanced across participants. **d** Time course of a single trial, composed of 4 different stages. Crucially, the ideation period is separated from the period in which participants had to report the story.

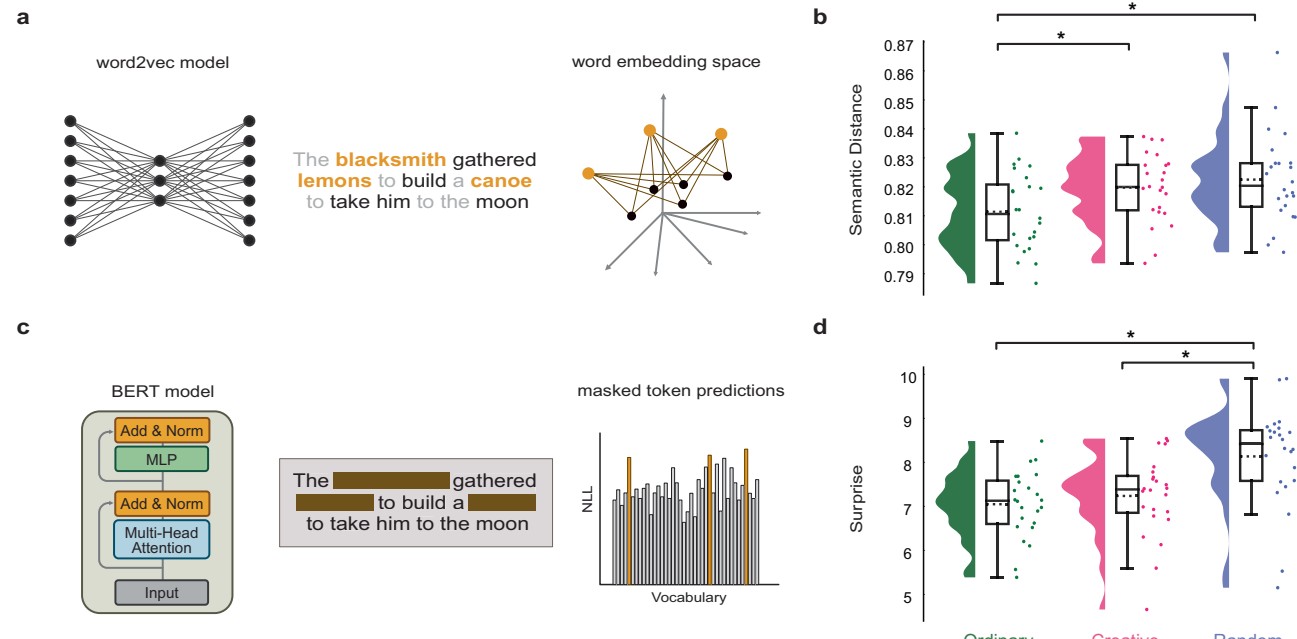

**Fig. 2 | Behavioural results using distributional semantics and large language models. a** Graphical representation of the pipeline adopted for the semantic distance analysis. We used word2vec models to extract word embedding vectors of the target (orange) and non-target (black) words from participants' stories, excluding stopping words (grey). The depicted story comes from a sampled participant from the creative condition. Then, we computed the semantic distance between target and non-target words and averaged the results across the 3 targets. **b** Raincloud plots depicting the semantic distance scores across Ordinary (green), Creative (pink), and Random (purple) conditions. Each dot represents a participant, horizontal bars with asterisk indicate statistical significance ($N = 24$) and dotted lines in the boxplot show mean values. **c** Graphical representation of the pipeline adopted for the surprise estimation analysis. We used large language models (BERT) to perform masked modelling by masking the target word tokens and computing the predictions to those masked tokens. We then computed the surprise score as the negative log-likelihood (NLL) between those predictions and the target words and averaged the results across the 3 targets. **d** Raincloud plots showing the surprise scores across the 3 conditions as in (**b**). Horizontal bars with an asterisk indicate statistical significance and dotted lines in the boxplot show mean values. For all boxplots in the raincloud plots, the dotted and solid horizontal line represent the mean and median value, respectively, while the whiskers extend to the minimum and maximum data point that does not exceed 1.5 times the interquartile range.

and colleagues[40] reported that functional connectivity networks in resting state predicted differentially novelty and appropriateness measures extracted from generated stories. Crucially, some of these studies are confounded by the motor artifacts derived by reporting the story[61], while others even lack direct measuring of the neural dynamics underlying the story ideation[40]. Among the studies which are unaffected by these limitations, only the neural activations are reported, i.e. the most active areas during the ideation of stories. Unfortunately, this approach is neglecting the intrinsic dynamic nature of ideating a story, leading to an incomplete characterization of how semantic control processes select and organize conceptual knowledge for generating narratives. Additionally, these studies did not adopt robust quantitative methods, such as deep language models, to assess the content of the generated stories and how linguistic features extracted from these stories are encoded in the neural dynamics. Importantly, none of these works compared the generation of these creative stories with proper control conditions, by selectively excluding the novelty and appropriateness components that characterize the ideation of creative stories. In sum, despite the high relevance of this topic for the neuroscientific understanding of language production and creative cognition, an extensive examination of the neural dynamics of generative storytelling is still missing.

Here, we sought to fill this gap by using functional magnetic resonance imaging (fMRI) to probe participants in a story generation task. We asked them to generate stories according to a set of instructions that modulate the level of semantic control requested for story generation, by separating the ideation period from the report one. Specifically, participants generated stories by controlling the amount of novelty and appropriateness in them, with creative stories regarded as the ones with a balance of these two features while ordinary and random stories lacked one of the two features[30,31,43,62]. We hypothesized that this experimental manipulation influenced the exploration strategies of participants when navigating the semantic space to generate a narrative. Specifically, we hypothesized that the exploratory behavior should be minimal in the ordinary condition, lacking novelty, and maximal in the random condition, lacking appropriateness, with the creative condition having an intermediate level of exploration. We leveraged recent deep language models[53,54] to investigate, at the behavioral level, how participants performed and whether novelty and appropriateness could be reliably extracted from linguistic stories using such computational approach. Thus, we investigated at the neural level what aspects of the neural dynamics could reliably distinguish creative stories from ordinary and random ones. First, we assessed neural synchrony across participants using inter-subject correlation (ISC)[63,64] to investigate individual differences in the neural activity underlying story ideation. Then, we employed multivariate pattern analysis (MVPA)[65] to investigate how patterns of brain activity differentiated the generation of creative stories from their counterparts lacking either novelty or appropriateness and analyzed how these features were encoded across brain regions. Moreover, we also decomposed the neural dynamics underlying the story generation into its connectome harmonics and investigated the same research questions, using the recently proposed framework of Connectome Harmonics Decomposition (CHD)[66–68], and specifically asking whether there are distinct spatial frequencies that discriminate the different ideation modalities or encode the linguistic features. In addition, we conducted functional connectivity analysis[69] to study how patterns of functional coupling within and between brain networks, known to be generally involved in creative ideation[41,42], differed when contrasting our experimental conditions. Finally, we investigated the temporal irreversibility of the neural dynamics to study the amount of non-equilibrium dynamics

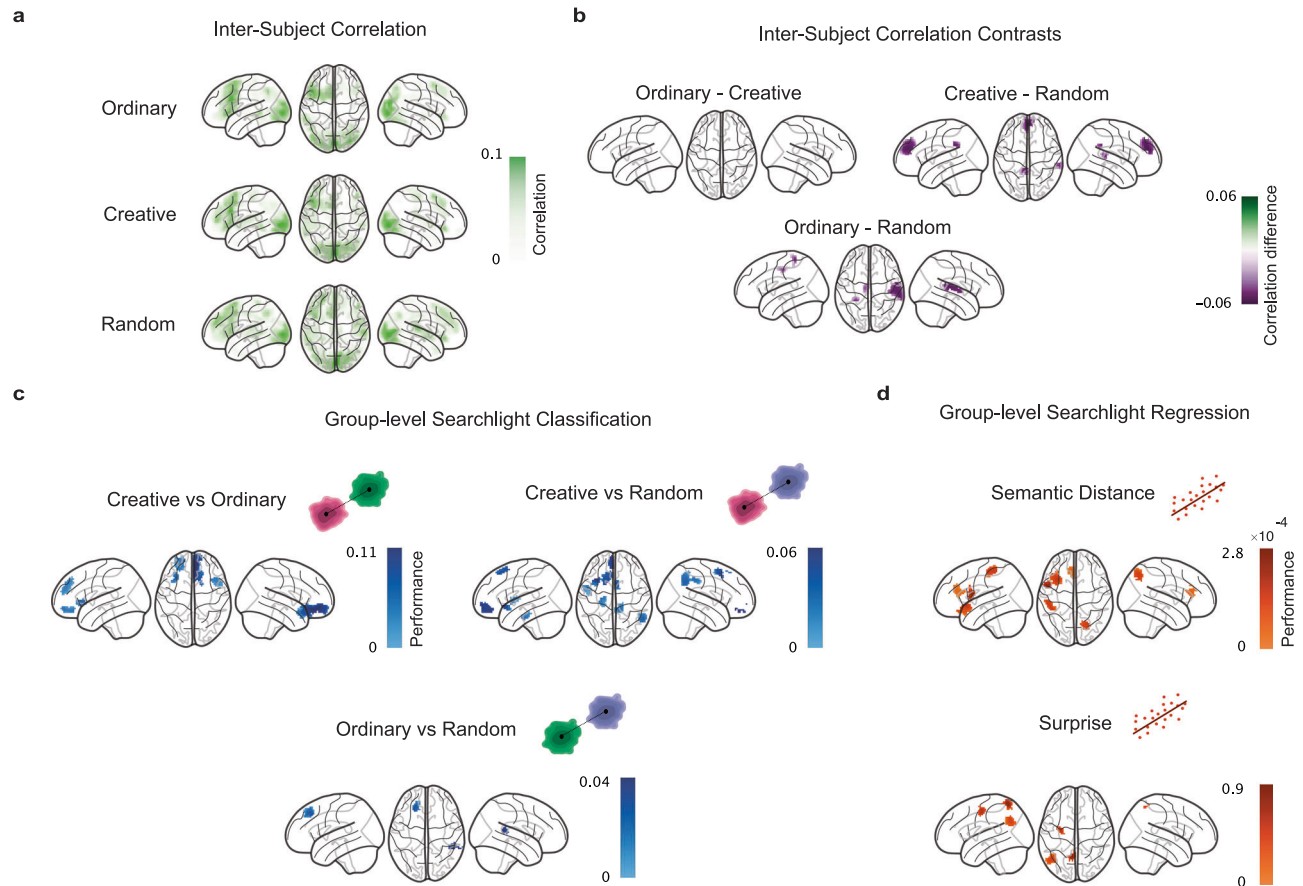

**Fig. 3 | Group-level inter-subject correlation and multivariate pattern analysis.** **a** Brainplots showing the inter-subject correlation results on individual conditions. Highlighted brain regions represent statistically significant areas. **b** Contrasts between conditions for the inter-subject correlations. Highlighted brain regions show statistically significant results. **c** Group-level searchlight classification between all pairs of conditions. Performance is computed as the difference between the classification accuracy and empirical chance level (50% expected in this case of binary classification). Highlighted brain regions in the glass brain plots showed statistically significant areas. **d** Group-level searchlight regression of the two behavioural features in the Creative condition. Performance is computed as the difference between the model $R^2$ scores and model's performance with permuted features. Highlighted brain regions in the glass brain plots showed statistically significant areas.

and entropy production underlying the story ideation, asking whether creative stories are characterized by an increased irreversibility compared to the other conditions.

## Results

Data from 24 human participants were collected by recording their fMRI signal while performing a story generation task (SGT). Our task consisted of generating a story with the constrain that 3 given target words must be included in the story. We manipulated the level of semantic control in the story generation process by having participants generate a story according to 3 different instructions[43], which we will refer to as the Ordinary (OR), Creative (CR), and Random (RA) condition. These conditions differed by the amount of 2 features, namely novelty and appropriateness, that were required to generate a story (Fig. 1a). In the OR condition, participants were instructed to maximize appropriateness and minimize novelty, while in the RA were instructed vice versa. In the CR condition, they had to attempt to maximize both features, resulting in a semantic control state that trades-off exploration and exploitation in the semantic space[31]. For a qualitative inspection of the generated stories, we sampled some of them having the same triplets of target words across participants and showed them in Fig. 1b. Participants performed 8 trials for each condition in the MRI scanner (Fig. 1c), resulting in a total of 24 stories per participant with target word triplets counterbalanced across participants. Crucially, we separated the period in which participants thought about the story (Ideation period) and the period in which they verbally reported the story (Report period), as

shown in Fig. 1d. This allowed us to avoid any possible motion and verbal confounds and clearly examine the neural dynamics of story generation[70]. After the data collection, we asked 2 independent human experts to transcribe the reported stories and started our data analysis pipeline from the behavioral data, to investigate whether and how participants conformed to the instruction set we provided. As a sanity check, we also asked 6 independent human raters to judge the generated stories from the whole sample across 3 scales corresponding to the same instruction sets (Figure S1a). Indeed, we found that the stories from the OR condition were evaluated as more ordinary than creative (t(23) = 4.77, $p_{FDR}$ = 0.0038, Cohen's d = 1.02, 95% CI [−0.18, 2.22]) and random (t(23) = 5.85, $p_{FDR}$ = 0.003, d = 2.71, 95% CI [1.15, 4.28]). We also observed that the CR stories were rated more creative than ordinary (t(23) = 8.42, $p_{FDR}$ = 0.0001, d = 1.51, 95% CI [0.23, 2.79]) or random (t(23) = 3.08, $p_{FDR}$ = 0.0135, d = 0.68, 95% CI [−0.48, 1.85]), and the RA stories were more random than creative (t(23) = 4.42, $p_{FDR}$ = 0.0053, d = 1.49, 95% CI [0.21, 2.77]) or ordinary (t(23) = 5.57, $p_{FDR}$ = 0.0045, d = 2.80, 95% CI [1.20, 4.39]).

Next, we leveraged recent advances in the natural language processing (NLP) field by adopting deep neural network (DNN) models trained on large corpora of text data to analyze our data[53,54]. The main reason why we made this methodological decision was due to the unique ability of these models to handle complex textual data such as stories[71,72]. We aimed to operationalize the two features (i.e., novelty and appropriateness) we asked participants to control in the story generation. Thus, we proposed a methodology based on a dual selection of DNN models and metrics that is

able to capture both features as distinctively and reliably as possible. For the novelty feature, we adopted the distributional semantics framework by using 3 word2vec models[31,53] to extract word embedding vectors from the stories (Fig. 2a). This approach allowed us to compute the semantic distance[31,43,73] metric between the target and non-target words (excluding stopping words). We utilized this measure as a proxy for how novel or original the story was, as it has been also similarly used to study divergent thinking[31,43,74]. For the appropriateness feature, we capitalized on current state-of-the-art transformer architectures for NLP[54,75], namely the Bidirectional Encoder Representations from Transformers (BERT) model. This model allowed us to mask the target words from each story and let the model predict what would have been the most appropriate word to fill the masked tokens (Fig. 2c). Thus, by computing from 2 BERT models the Negative Log-Likelihood (NLL) between the actual target and the model predictions we obtained a measure of how much appropriate that story was, a measure that we referred to as surprise (since technically it can be seen as the Shannon surprise). When we applied these two measures to our data, we expected to find an increased semantic distance and surprise from the OR to the RA condition, with the CR standing in the middle. Crucially, we found that the semantic distance (Fig. 2b) was significantly higher in the CR compared to OR ($t(23) = 3.09$, $p_{FDR} = 0.0075$, $d = 0.65$, 95% CI [0.07, 1.23]) and higher in RA than in OR ($t(23) = 2.59$, $p_{FDR} = 0.012$, $d = 0.76$, 95% CI [0.18, 1.35]). No significant difference was observed between CR and RA ($t(23) = 0.83$, $p_{FDR} = 0.209$, $d = 0.20$, 95% CI [-0.36, 0.77]). We also found that the surprise (Fig. 2D) was significantly higher in the RA compared to CR ($t(23) = 2.74$, $p_{FDR} = 0.009$, $d = 0.87$, 95% CI [0.28, 1.46]) and higher in RA than in OR ($t(23) = 3.90$, $p_{FDR} = 0.0015$, $d = 1.13$, 95% CI [0.52, 1.74]). No significant difference was observed between CR and OR ($t(23) = 0.78$, $p_{FDR} = 0.221$, $d = 0.22$, 95% CI [-0.35, 0.79]). We also ran additional analyses on the individual models, since our measures were an average over multiple models (3 word2vec and 2 BERT) for improving the methodological robustness, and found that the results replicated as well (Fig. S1b-c). Finally, we ran a correlation analysis between the semantic distance and surprise measures to confirm that these two measures captured different aspects of the data and found that indeed the correlation coefficient was low ($r = 0.234$, Figure S1d). Thus, our results evidenced how participants indeed controlled the level of novelty and appropriateness in the CR condition.

After having established that, at the behavioral level, participants followed our instructions and correctly exerted the right level of semantic control, we investigated the neural mechanisms underlying these behavioral effects. We conducted all the neural analyses only on the entire ideation period, given that a crucial aspect of our design was to measure the neural dynamics underlying narrative generation in this period free of any motion artifacts. Thus, to confirm that using the BOLD signal in the whole ideation period was capturing relevant brain activity, we computed the variance of the signal across trials (stories) along each volume. We found that indeed, in the later stages of the ideation period, the variance decreased in all conditions and in both the whole brain (Figure S2a) and all the 7 regions of interest (Figure S2b) taken from the Yeo atlas[76]. A reduction in variability typically indicates that participants converge on similar processing, which in our case presumably indicates returning to a resting or preparatory state before providing their narrative. Therefore, we started analyzing the neural data by assessing the similarity of the neural dynamics underlying the generation of the stories by means of inter-subject correlation (ISC) analysis[63,64]. In essence, we asked which brain areas had a similar temporal profile in the ideation period across participants. Crucially, we computed ISC for each condition by separating the stories having the same triplet of target words and averaging the results across target word triplets. This helped us to directly quantify the degree of similarity across participants when ideating a story starting from the same linguistic constrains. Interestingly, we found that for all conditions (Fig. 3a), the left frontal cortex (all $p_{FDR} < 0.002$, peak ISC values for Ordinary is 0.187, Creative is 0.123, and Random is 0.176) and the visual cortex (all $p_{FDR} < 0.003$, peak ISC values for Ordinary is 0.144, Creative is 0.132, and Random is 0.148) were significantly similarly activated across participants. Moreover, when contrasting the ISC values across

conditions (Fig. 3b), we found that the random stories had a significantly more similar underlying neural dynamics across participants compared to both ordinary, in bilateral parietal cortices (all $p_{FDR} < 0.04$), and creative stories, in parietal and frontal cortices (all $p_{FDR} < 0.02$). In sum, our findings highlight that during narrative generation, left frontal and visual cortices consistently showed inter-subject similarity across all conditions, while random story ideation elicited even higher alignment in parietal–frontal areas, underscoring distinct patterns of neural synchronization in response to different storytelling constraints.

Next, we employed Multivariate Pattern Analysis (MVPA)[65] methods to extract multivariate patterns of neural activity in the ideation period that differentiated how participants exerted their semantic control across conditions. We performed all the analyses using the blood-oxygen-level-dependent (BOLD) signal, instead of using a conventional general linear model (GLM) approach. We opted for this since our ideation period lasted for 30 seconds and we considered it unreasonable to search for voxel activations for such a long time period using a GLM. Critically, we performed searchlight analyses[77] at the group-level to control for the effects of the target words that were counterbalanced across participants and to identify population-wise invariants from neural activity patterns[78]. First, we conducted classification analyses between every pair of conditions (Fig. 3c) and computed the classifier's performance as the difference between the classifier's accuracy and the accuracy of a model with permuted labels. In other words, a performance of 0 corresponds approximately to a classification accuracy of 50% since we always performed binary classification analyses. We also performed statistical analysis on the searchlight results using false discovery rate (FDR) by setting an alpha level of 0.01 for reporting in the main text, but we also reported the results thresholded with an alpha level of 0.05 in Figure S3. When classifying CR versus OR, we found a cluster of frontal regions, such as the prefrontal cortex, being significantly different between conditions (Fig. 3c; PFC, performance=0.06, $p_{FDR} = 0.001$). Specifically, the most prominent effects were located at the orbitofrontal cortex (OFC, performance=0.10, $p_{FDR} = 0.0098$), the dorsomedial prefrontal cortex (dmPFC, performance=0.13, $p_{FDR} = 0.0095$) and the left frontal operculum (LFO, performance=0.09, $p_{FDR} = 0.0031$). Then, we classified CR against RA and found that a fronto-temporo-parietal network of brain areas significantly differentiated their neural activations. Here, the most prominent effects were located at the PFC (performance=0.06, $p_{FDR} = 0.0065$), the LFO (performance=0.01, $p_{FDR} = 0.0067$), the medial parietal cortex (MPC, performance=0.05, $p_{FDR} = 0.0066$) and the left temporal pole (LTP, performance=0.03, $p_{FDR} = 0.0066$). Moreover, classification between OR and RA revealed significant activity in PFC (performance=0.03, $p_{FDR} = 0.0067$) and right temporal occipital parietal cortex (performance=0.04, $p_{FDR} = 0.0095$). Then, we also asked ourselves how the semantic distance and surprise features extracted from the generated creative stories were mapped onto the cerebral cortex (Fig. 3d). In other words, we asked which brain areas encoded these features in the CR condition that most probably gave rise to the observed behavioral outcome. To answer this question, we performed searchlight regression analyses on each feature and measured the model's performance as the difference between the model's $R^2$ score and the $R^2$ of a model with a randomized dependent variable, with an alpha level set to 0.05 for FDR correction. Note that, here the models' performance scores were affected by the scale of each feature. This analysis revealed that the semantic distance was encoded most prominently in a fronto-parietal network comprising the PFC (performance=$2.33 \times 10^{-4}$, $p_{FDR} = 0.0329$), the LFO (performance=$3.41 \times 10^{-4}$, $p_{FDR} = 0.0309$) and the posterior parietal cortex (PPC, performance=$1.43 \times 10^{-4}$, $p_{FDR} = 0.0296$), while the surprise was encoded mostly in the PPC (performance=1.36, $p_{FDR} = 0.0164$) and the frontal eyes fields cortex (FEF, performance=1.06, $p_{FDR} = 0.046$). All together, these results showed how, at the neural level, different activation patterns underlay the participants' semantic control strategies and supported the results at the behavioral level.

Thus, after having identified specific brain regions exhibiting significant activity through the searchlight analysis, we next extended our investigation to a more system-level, whole-brain approach using

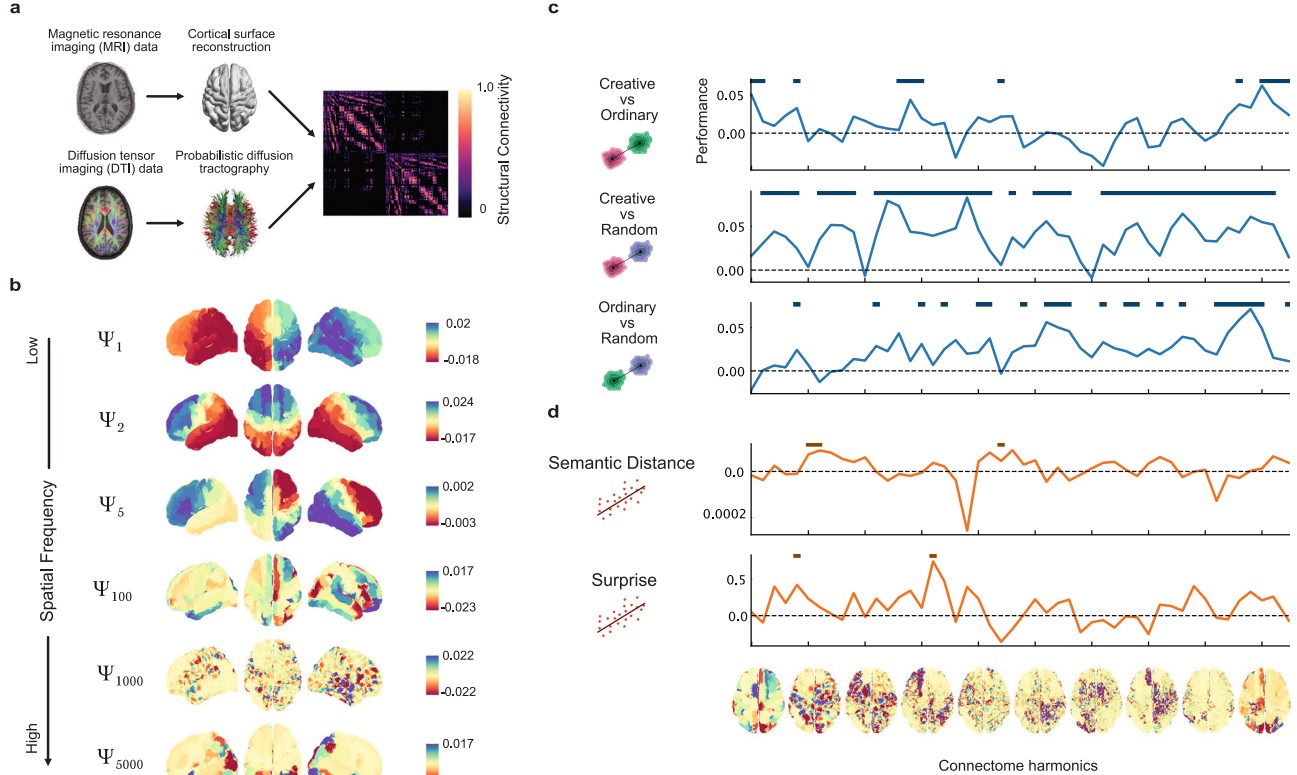

**Fig. 4 | Connectome Harmonic Decomposition reveals spatial frequency patterns underlying the modulation of semantic control during story generation. a** Data analysis pipeline for implementing the Connectome Harmonic Decomposition. **b** Brain plots showing examples of harmonics. **c** Group-level searchlight classification across the harmonic space between all pairs of conditions. Performance is computed as the difference between the classification accuracy and empirical chance level (50% as the expected value). Horizontal lines indicate statistical significance. **d** Group-level searchlight regression across the harmonic space for both behavioral features. Performance is computed as the difference between the model $R^2$ scores and model's performance with permuted features. Horizontal lines indicate statistical significance. Brain plots across the x-axis depict the median harmonic inside each bin.

Connectome Harmonic Decomposition (CHD, Fig. 4a)[66]. While the searchlight analysis provided valuable insights into localized brain regions involved in semantic control for generative storytelling, CHD allowed us to study how the entire brain organizes itself to explore the semantic space and generate a story coherently with the instruction sets. CHD, based on the mathematical framework of graph Fourier transform, decomposes neural signals into a set of harmonic modes (Fig. 4b), each representing different spatial patterns of brain activity[68]. This approach is grounded in the idea that brain functions can be understood as a complex orchestration of these harmonics, analogous to musical notes forming a symphony, offering a unique perspective on the brain's intrinsic architecture and its role in shaping neural dynamics[66]. It consists of the eigen decomposition of the structural connectivity matrix containing both local and global information taken from diffusion tensor imaging (DTI) and cortical surface reconstruction data (Fig. 4a). The resulting harmonic modes are then arranged from low to high spatial frequency, where low frequency means that network proximity plays a huge role in the signal values (Fig. 4b). We then applied the CHD to our fMRI data and performed classification analyses between conditions on the binned (48 bins) harmonic energy values across the spatial frequency dimension (Fig. 4c). We found that a specific pattern of low, medium and high spatial frequencies significantly discriminated the CR condition from the OR, with the peak of performance being in the high frequency range (bin 45, performance=0.06, $p_{FDR} = 0.0106$). When classifying CR against RA, we found a different pattern of spatial frequencies and a general higher discriminability with respect to the CR versus OR, with the peak of performance being in the low-medium frequency range (bin 19, performance=0.08, $p_{FDR} = 0.009$). We also found the OR and RA were significantly classified by another frequency pattern, especially in the high frequency range (bin 44, performance=0.07, $p_{FDR} = 0.0229$). Analogously to

the searchlight approach, we also applied regression analyses to investigate how semantic distance and surprise were encoded at the whole-brain level in the spatial frequency dimension (Fig. 4d). We found that semantic distance was encoded specifically in the low and medium frequencies (peak at bin 6, performance=$1.01 \times 10^{-4}$, $p_{FDR} = 0.0372$), and similarly was encoded the surprise in that spatial frequency range (peak at bin 16, performance=0.74, $p_{FDR} = 0.0223$). Taken together, these findings demonstrated that the neural dynamics of the whole-brain is organized in specific spatial frequency patterns, representing the decoupling of functional neural activity from the underlying structural connectivity, during the ideation of complex linguistic outcomes such as stories requiring different abstract features to be incorporated.

Then, we turned our attention to the direct functional relationships between brain regions with a hypothesis-driven functional connectivity (FC) analysis[69]. While CHD provided a window into the global orchestration of brain activity, functional connectivity analysis allowed us to map and quantify the strength and nature of the associations between specific brain areas. Based on previous results in the literature about semantic control[17,41] and our results from the searchlight analysis, we selected and analyzed the functional connectivity within and between the brain areas of the following brain networks (Fig. 5a): the Default Mode Network (DMN), the Salience Attention Network (SAN) and the Frontoparietal Control Network (FCN). We found that the average FC within the DMN (i.e., functional connectivity between brain areas belonging to the DMN) was significantly higher in CR (Fig. 5b) compared to OR (t(23) = 2.58, $p_{FDR} = 0.0255$, $d = 0.62$, 95% CI [0.04, 1.2]), with stronger connections encompassing the precuneus with left frontal areas. No difference was observed between CR and RA (t(23) = 1.00, $p_{FDR} = 0.166$, $d = 0.20$, 95% CI [-0.36, 0.77]) and OR and RA (t(23) = 1.57, $p_{FDR} = 0.096$, $d = 0.36$, 95% CI [-0.20, 0.94]). We also found that the average

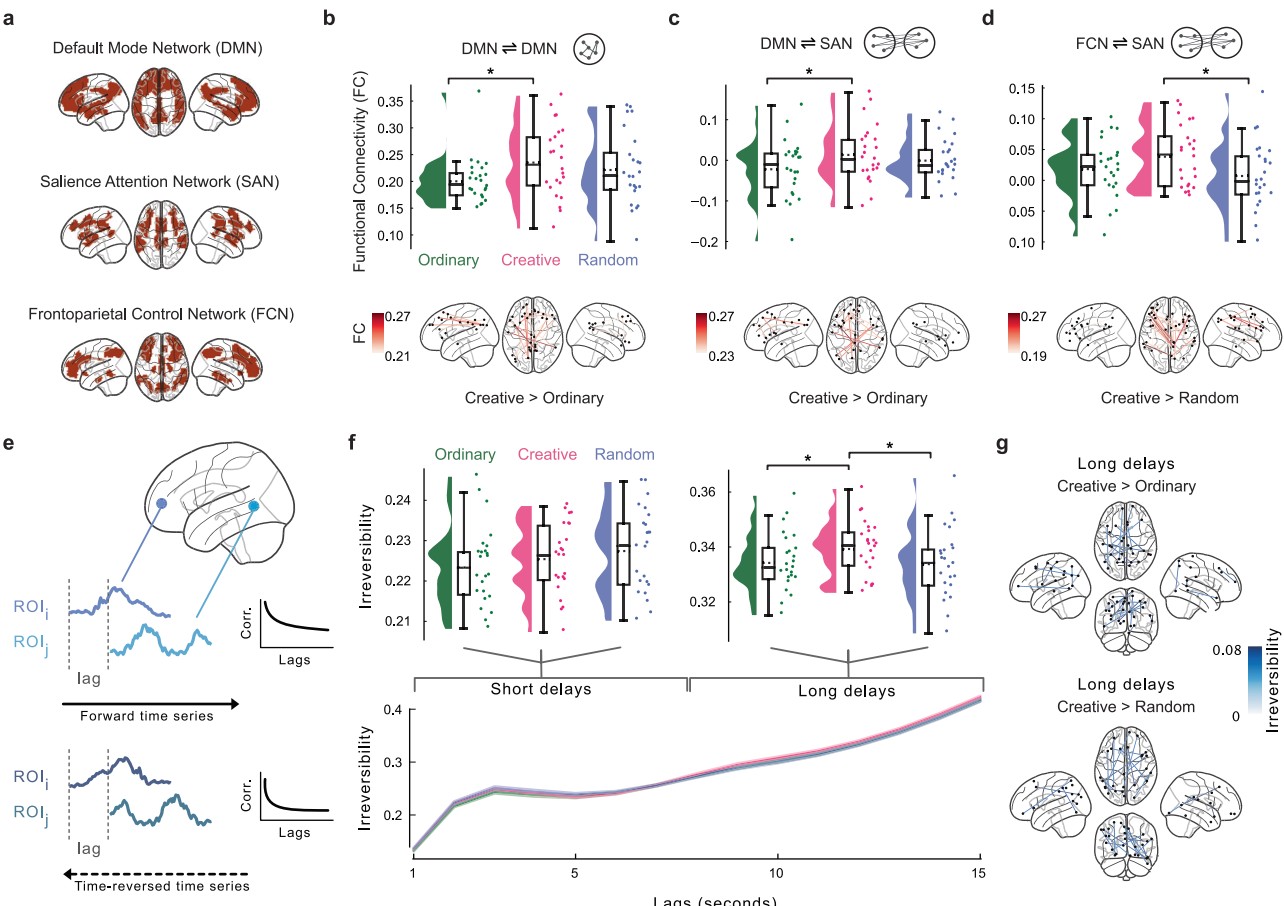

**Fig. 5 | Functional connectivity among brain networks and temporal irreversibility of neural dynamics underlying story generation. a** Cortical distribution of the selected brain networks, namely the Default Mode Network (DMN), the Salience Attention Network (SAN) and the Frontoparietal Control Network (FCN). **b** Top, raincloud plots showing within DMN functional connectivity averaged across all connections for each condition. Each dot represents a participant, while horizontal bars with an asterisk indicate statistical significance ($N = 24$). Bottom, connectivity brain plot showing the top 5% connections within DMN when contrasting creative and ordinary stories. In (**c**), the same plot as in (**b**), but showing the functional connectivity between the DMN and SAN. Also in (**d**), functional connectivity between the FCN and SAN, with the bottom contrasting creative and random stories. **e** Schematic representation of the method used to quantify the irreversibility

of the neural dynamics from the time-lagged directed connectivity analysis. The purple and azure time-series represent an example of two ROIs in the forward and reverse mode. On the right, the cross correlogram as a function of the delays. **f** On the bottom, lineplots showing the irreversibility values as a function of the lags in the three conditions. Top, raincloud plots indicating the difference between conditions for short and long delays (lags). Each dot represents a participant, horizontal bars with an asterisk indicate statistical significance ($N = 24$). **g** Connectivity brain plot showing the significant ($p < 0.0001$, uncorrected) directed connections when contrasting creative with ordinary and random stories. For all boxplots in the raincloud plots, the dotted and solid horizontal line represent the mean and median value, respectively, while the whiskers extend to the minimum and maximum data point that does not exceed 1.5 times the interquartile range.

---

FC between DMN and SAN (Fig. 5c) was higher in CR compared to OR ($t(23) = 2.35$, $p_{FDR} = 0.042$, $d = 0.50$, 95% CI [0.08, 1.07]) but not compared to RA ($t(23) = 0.93$, $p_{FDR} = 0.182$, $d = 0.22$, 95% CI [-0.34, 0.79]). Also here, posterior parietal regions were mostly coupled with left frontal areas in CR compared to OR. No difference was observed between OR and RA ($t(23) = 1.45$, $p_{FDR} = 0.120$, $d = 0.34$, 95% CI [-0.23, 0.91]). Moreover, we observed that the average FC between FCN and SAN (Fig. 5d) was higher in CR compared to RA ($t(23) = 2.45$, $p_{FDR} = 0.033$, $d = 0.61$, 95% CI [0.03, 1.19]), with stronger connections between parietal areas with bilateral frontal regions, but not compared to OR ($t(23) = 1.69$, $p_{FDR} = 0.078$, $d = 0.42$, 95% CI [-0.15, 0.99]). No difference was observed between OR and RA ($t(23) = 0.88$, $p_{FDR} = 0.1935$, $d = 0.22$, 95% CI [-0.35, 0.79]). Also, no difference was observed in all other within and between networks comparison that we reported in Figure S4. Overall, our findings highlight the critical neural pathways for semantic exploration during the ideation of stories and contribute to a deeper understanding of the network dynamics underpinning generative storytelling.

Finally, we complement our previous FC analysis by investigating the temporal irreversibility of the neural dynamics underlying the ideation

period. Essentially, we quantified the degree of temporal asymmetry in the directed connectivity, measured as time-lagged correlations between pairs of ROIs (Fig. 5e). This provided a measure of non-equilibrium dynamics and entropy production. After computing the irreversibility matrix for each condition at different lags, we plotted the average across all connections as a function of lags (Fig. 5f). We found that, while at shorter delays in the order of few seconds (1–7 seconds) there was not a clear distinction among the conditions, at longer delays (8–15 seconds) the creative condition exhibited a consistently level of temporal irreversibility that was higher than the other conditions. Specifically, when we averaged these results on the long delays (Fig. 5f top), we observed a significantly higher irreversibility in creative stories than in ordinary ($t(23) = 2.59$, $p_{FDR} = 0.024$, $d = 0.50$, 95% CI [-0.07, 1.07]) and random ($t(23) = 2.06$, $p_{FDR} = 0.038$, $d = 0.49$, 95% CI [-0.08, 1.06]) ones. No significant difference was observed at shorter delays (all $p_{FDR} > 0.05$). Thus, we inspected what were the directed asymmetrical connections that drove the results on the long delays. By contrasting the creative condition with the ordinary and the random, we found a large-scale network involving mostly occipital, parietal and frontal regions for both contrasts (Fig. 5g, all $p < 0.0001$, uncorrected). Together, these findings

uncover the neural dynamical signatures that distinguish creative stories from ordinary or random ones in terms of temporal irreversibility and non-equilibrium dynamics.

## Discussion

In the present study, we investigated the neural dynamics underlying generative storytelling by varying the demands of semantic control within the narrative ideation. We employed a within-subjects design, tasking participants with creating narratives under three distinct instruction sets, each varying in degrees of novelty and appropriateness. Our paradigm allowed us to address several research questions about how semantic control operates in naturalistic conditions such as free narrative production and its neural signatures.

Crucially, our experimental design allowed us to disentangle the period in which participants ideated the narrative from when they reported it. This design choice was not adopted by all previous studies on generative storytelling[40,58,60,61] and, thus, provided us with a clean time window characterized only by the semantic control processes operating to generate stories according to the instruction, free from motion and vocalization artifacts. Another critical feature of our study design is that we control for the inclusion of both novelty and appropriateness features[30,31,43,62], while often only one of these features is controlled especially in the context of naturalistic narrative production[40,58,60,61].

We started our analysis plan by first adopting a computational approach to operationalize the abstract features we required participants to manipulate during the generation of the stories. Our approach relied on recent advances in natural language processing based on deep neural network models that learn statistical patterns from large text corpora[53,54,71,79] and allowed us to successfully differentiate the novel and appropriate components of complex textual data such as narratives. Note that, although the use of semantic distance as a measure of novelty or originality has been extensively used in previous research[31,43,74], our measure of appropriateness, derived from the BERT model's surprise, has not been used yet and in our opinion can be a valid measure for further studies attempting at quantifying the usefulness of a narrative in the creative cognition field.

Thus, after we observed at the behavioral level that participants controlled the level of novelty and appropriateness in the generation process of creative stories, we assessed how similar was the neural dynamics across participants when ideating a story by means of the ISC analysis[63,64]. Notably, two primary cortical regions, the left frontal cortex and the visual cortex, demonstrated significant inter-subject similarity across all conditions, irrespective of the level of semantic control. These findings suggest that these regions are central cortical hubs in narrative generation. This is particularly compelling given that no visual output was required during story ideation, underscoring the intrinsic role of the visual cortex in constructing internal representations during story generation. This aligns with Erhard et al.[60], who found visual cortex involvement in expert writers. However, our results show this engagement across all participants and conditions, suggesting that narrative ideation inherently relies on visual processing, likely due to the mental imagery involved in constructing coherent stories[80]. Interestingly, our analysis revealed that random stories elicited greater intersubject neural synchrony in frontal and parietal areas than both creative and ordinary stories. One possible explanation is that random story generation, by imposing minimal semantic control, induces a more uniform or stereotypical response in these high-level regions, i.e. participants essentially aligned on a simpler pattern of activity when free from semantic constraints. Another possibility is that this minimal semantic control reduces the variability of cortical engagement, leading to higher similarity across individuals.

Next, we performed multivariate decoding analyses to investigate which brain areas differed among the creative, ordinary, and random generated narratives based on their activity pattern in a searchlight fashion[77]. We found that creative stories were distinguished from ordinary stories mostly by the neural activity located in the dmPFC, LFO, and OFC, while from random ones by the activity pattern in PFC, LFO, LTP, and MPC. We

interpreted these findings as evidence that different semantic control strategies were implemented in the generation of these different types of stories. Particularly, the involvement of the orbitofrontal cortex and dmPFC in the former comparison could be the result of the increased exploratory behavior in the semantic space required to generate creative stories[31,81], given that the instructions set a higher value of novelty for this condition compared to the ordinary one. These areas have been predominantly indicated in the literature as crucial cortical hubs for the pursuit of exploration in decision-making and the resolution of the exploration-exploitation trade-off in reinforcement learning[82-85]. Also, the involvement of the medial parietal cortex in the latter comparison could be ascribed to the higher semantic constrains imposed in creative story generation, since this region is known to be important for memory retrieval and attentional control[17,86,87]. Interestingly, our observation of parietal involvement diverges from a recent meta-analysis[20] that found no parietal activation for semantic control. One likely explanation is that the meta-analysis primarily examined simpler, word-level or passive comprehension tasks, whereas our study employed an active, more complex narrative-generation paradigm. We speculate that this increased complexity may specifically recruit parietal regions that do not come into play when the task is limited to simpler or more passive forms of language processing. Importantly, we observed the activity patterns in the LFO and PFC being different in both comparisons, indicating a unique control strategy to access semantic representations when participants balanced novelty and appropriateness[17,25,88]. Thus, our results confirm and extend previous studies to complex and naturalistic settings such as generative storytelling, since all the brain areas we found in our study were also reported in the semantic cognition literature[17].

Moreover, we also tried to directly predict the trial-by-trial fluctuations of semantic distance and surprise during creative story generation from the neural activity to investigate where in the brain these features were mostly encoded. Strikingly, we observed that the semantic distance was encoded predominantly in fronto-linguistic areas such as LFO, while the surprise was encoded in parietal areas such as PPC. We interpreted these findings as crucial confirmation of the differences we observed between conditions at the behavioral level since these areas largely overlapped with the comparison results we obtained at the neural level in the searchlight analysis. In other words, the semantic distance feature that was higher in creative compared to ordinary stories was encoded in brain areas that belonged to the set of areas that differentiated the neural activity underlying the ideation period of these two conditions. Similarly, we found parietal cortices encoding the surprise feature related to the areas resulting from the searchlight classification of the creative versus random stories. Note that, our semantic distance measure is a generalization to narratives of the usual measure applied in previous studies to pairs of words[31,43,73]. Yet, we found surprisingly similar areas encoding this measure, extending previous results from word pairs to complex sentence level structures[17]. It is also noteworthy that our measure of surprise suited for long text data was better predicted by parietal areas that are found in the literature to be involved in the computation of surprise in decision-making tasks[89].

Next, we focused our investigation on a more system-level, whole-brain approach using Connectome Harmonic Decomposition (CHD)[66], which consists of decomposing the neural activity, via the structural connectivity matrix, into its connectome harmonic modes ranging from low to high spatial frequency patterns. CHD offers a foundational shift from the traditional perspective that views brain activity as consisting of isolated spatial units, presenting additional and complementary insights that the location-focused approach does not capture[68]. We found that low-medium spatial frequency patterns discriminated mostly creative from ordinary story generation, while high frequency connectome harmonics differentiated creative from random stories. Also, we observed how the features representing novelty and appropriateness of the stories were encoded in low-medium frequency connectome harmonics. Generally, we showed how the neural dynamics at the whole-brain is arranged in specific frequency patterns of connectome harmonics during story ideation requiring the modulation of different abstract features. Note that, the transition from low

to high spatial frequencies can be interpreted as a growing decoupling of functional neural activity from the underlying structural connectivity[90,91]. Therefore, the CHD framework allowed us to reveal the multi-scale network organization of the connectome as well as the asymmetric relationship between brain structure and function when participants performed naturalistic story generation[68].

Then, we turned our investigation to study functional connectivity among the brain areas we found previously. Based also on previous results in the literature about semantic cognition[17,41], we identified and selected the DMN, SAN, and FCN and compared the within and between functional coupling among the conditions. Recent evidence has highlighted the importance of these large-scale brain networks for domain-general problem-solving and their causal role in information processing during task-related decision-making[92]. We observed creative story generation being characterized by an increased FC within the DMN and between the DMN and SAN compared to ordinary generation and by increased coupling between FCN and SAN compared to random generation. These findings highlight previous findings on the role of the default network being implicated in spontaneous associations[41,93,94] and control network being related to attentional inhibition[95,96]. Interestingly, our results suggest that the salience network may be responsible for orchestrating the activity of the default and control network when the balance between novelty and appropriateness is required, which falls well in line with previous research in this direction[41,92,97].

Finally, we investigated the temporal irreversibility of the neural dynamics underlying the story ideation. We achieved this by observing the temporal asymmetry between directed functional couplings among brain areas[98]. The more asymmetrical is the lagged interaction between two areas in the temporal domain, the more the system is irreversible. Conversely, if there is no difference between the temporal sequence of brain states in forward or reverse mode, then the system is reversible. Crucially, this temporal irreversibility can be seen as a marker of directed information flow between brain regions, with higher irreversibility signaling a more pronounced hierarchy (more directed interactions) of causal interactions in the neural dynamics. Remarkably, we found that creative stories required a higher level of irreversibility compared to both ordinary and random ones. In other words, there was a higher level of hierarchical processing among critical cortical hubs, which comprised linguistic, visual and high-level areas, during the ideation of a creative story. This could be explained by the observation that creative stories required generally more effort than the control ones, since the need to balance two essential features as novelty and appropriateness, which is reflected in the higher need to orchestrate and structure the neural dynamics to reach the creative solution. Notably, we observed this effect at long delays in the order of 8 to 15 seconds. We reasoned that this is line with the intrinsic time scale of the generative task, since generating a story requires much more recurrent cognitive dynamics than simple word associations, which can span several seconds. Moreover, it is also important to note that the notion of temporal irreversibility is closely tied to the concept of entropy production, which is not equivalent to entropy. Entropy measures the degree of disorder in a system, and it reaches its maximum at thermodynamic equilibrium, where the system has diffused into the largest possible number of states. At this point, entropy production becomes zero because the system no longer changes its distribution of states. In order words, if the entropy of the system is maximum, there is no chance to increase it. At thermodynamic equilibrium, the probability distribution over the system's microstates becomes stationary, even though individual state transitions continue, the overall distribution remains unchanged, resulting in zero net entropy production. Once the system is driven away from equilibrium, the absence of detailed balance leads to entropy production and marks the emergence of temporal irreversibility. We highlighted this distinction since there is reported in the literature evidence for increased resting-state entropic brain dynamics in divergent thinking[99]. Our irreversibility results contrast these previous findings, since higher irreversibility are linked to lower entropic states. We propose three main explanations for this discrepancy. First, the earlier work relied on correlating

resting-state measures with behavioral performance rather than directly assessing the entropy of neural dynamics during creative ideation, as we do by focusing on the ideation period itself. Second, the reported findings were primarily regional (mostly localized to left frontal areas), whereas our results capture network-level dynamics across the brain. Third, the task in that prior study focused exclusively on novelty, whereas our paradigm addresses both novelty and appropriateness, thus encompassing the full range of creative cognition processes. In sum, to our knowledge, our study is the first to establish the level of entropy production, and indirectly the entropy, of the neural dynamics at the network level during the ideation of creative narratives.

Building on our findings, these offer promising avenues for investigating how narrative generation processes may be altered in clinical populations where language or semantic memory function is compromised, such as individuals with schizophrenia or other disorders affecting semantic control. Tracking the neural signatures of generative storytelling identified here could shed new light on the cognitive and neural disruptions that underlie clinical language and high-level cognitive processing deficits. Likewise, examining how these findings about the neural dynamics of narrative production are also present in children could reveal important milestones in the maturation of semantic control.

In conclusion, our results highlight the neural mechanisms underlying the regulation of semantic exploration during naturalistic narrative production and contribute to a deeper understanding of the neural mechanisms underlying the role of semantic control in generative storytelling.

## Methods
### Participants
The initial sample included twenty-five individuals who were recruited from the University of Trento through advertisements in online platforms. None of the volunteers had a history of psychiatric or neurological disorders, were on psychoactive drugs, or had any MRI-related contraindications. Of the 25 volunteers, one was removed because of technical issues during the recording of behavioral data. Thus, the final sample used for the analysis included 24 healthy individuals (50% female), Italian language speakers, aged between 19 and 33 years (M = 25.67, SD = 4.48). The study protocol (2020–018) was approved by the Human Research Ethics Committee of the University of Trento. All ethical regulations relevant to human research participants were followed. Participants provided their informed consent and were compensated for their participation.

### Stimuli
The word stimuli were extracted from a starting pool of 60 concrete words from 6 categories (plants, jobs, objects, animals, places, vehicle). The length of the words was controlled, ranging from 3 to 8 letters. Since the word stimuli were presented in triplets, we computed all possible combinations of words with the constrain that a triplet can contain maximum one word per category. Then, we computed the semantic similarity for each pair of words inside a triplet (word 1 - word 2; word 1 - word 3; word 2 – word 3), using a word2vec model architecture called Continuous Bag Of Words (CBOW)[100]. The CBOW model is a deep neural network that is trained to predict a target word according to its context. This model was trained on Wikipedia articles and the OSCAR multilingual corpus[101] and its output is a 300-dimensional vector. To compute the semantic similarity, we calculated the cosine distance between the two vectors associated with the considered words. Also, we extracted the word frequency of each word based on the PAISÀ corpus[102], which comprises about 20 M tokens, mainly including Wikipedia pages and, to a minor extent, Wikibooks and Wikinews documents. Frequency values were normalized using the minmax scaler in the range of 0 and 1. Once we had semantic similarity and frequency values for all the triplets, our goal was to select the 24 triplets for the experiment based on the following criteria. First, the semantic similarity of each of the 3 pairs of words has to be somehow in the middle, in order to avoid triplets of words that are semantically too close or too distant from each other. Therefore, we set the ideal value of the semantic similarity to 0, since this measure ranges from -1

to 1. Second, as for the semantic similarity, the frequency value has to be neither too high nor too low, therefore we set the ideal value to 0.5. Since we have to select these triplets based on 6 values (3 semantic similarity pair values, 3 word frequency values) we use the Mean Square Error (MSE) to quantify how close these values are to the ideal target values. The final set of stimuli used in the experiment consists of the 18 triplets that have the lowest MSE value and do not contain words that have been already selected in other triplets. For an extensive description of the cued words listed in the Supplementary Materials, see Table S1.

## Procedure

Participants were instructed to perform a story generation task (SGT). A set of three target words was presented to participants and they were asked to think of a plot of a story that included these words and then vocalize it. Word triplets were controlled for possible semantic confounds (e.g. semantic similarity, length in letters, avoiding category repetition etc.) and grouped in three sets of 8 word triplets each. The experiment consisted of three conditions where individuals performed the SGT according to different instructions[31,43]. For the ordinary condition, the instructions were: "When you see three words, use them all to formulate an appropriate, unoriginal story. By 'ordinary' we mean a coherent, sensible, common story that could happen often. A story that would probably come to mind for many people when they read the words presented.". For the random condition, the instructions were: "When you see three words, use them all to formulate an inappropriate and original story. By 'Random' we mean an unusual, non-sensical and infrequent story. A story that would not occur mind of any other person when reading the words presented.". For the creative condition, the instructions were: "When you see three words, use them all to formulate an appropriate and original story. By 'creative' we mean a story that is coherent and makes sense, but in an unusual and infrequent way. A story that is neither ordinary nor random and that would come to mind for few other people when read the words presented.". We also presented an exemplar story for each condition based on a triplet word that was not included in the task. Participants completed the SGT in three runs of a block design. The order of conditions and the assignment of word triplet sets to each condition were counterbalanced across participants. While the order of triplet presentation within each condition was randomized across participants, the order of words within a triplet remained fixed. Each of the three blocks consisted of 8 trials resulting in a total of 24 stories per participant. Before starting the experiment, in order to familiarize themselves with the task, participants performed 2 practice trials using words that did not appear in the word stimuli used in the experiment. The time-course of a single trial (90 s) is illustrated in Fig. 1d. Each trial began with a fixation-cross in the middle of a black screen as a baseline for 25 sec. Then, the word triplet was shown for 5 sec. When the words disappeared from the screen, a fixation-cross appeared in the middle of a black screen indicating the beginning of the ideation period, in which participants had to generate the story in their mind for at most 30 seconds. Crucially, during this time period, no response was required from participants. Afterwards, the image of a microphone presented on the screen signaled participants to verbally report the generated story. They were instructed to report the story without further elaboration from the ideation period. We used an MRI compatible microphone to record the participants' responses. This fixed-timing approach ensures that the number of trial repetitions and time spent on the story ideation is equal across participants, which implies higher experimental control enabling a more straightforward analysis of the brain data. The experimental procedure was implemented using Opensesame[103] with PsychoPy as the backend[104].

## Behavioral data analysis

We started our analysis of the behavioral data by enrolling 2 independent human experts in the Italian language to faithfully transcribe the reported stories into text. Only one story from one participant was not possible to decode from the audio recording and this trial was excluded from the behavioral analyses as well as from all the subsequent neural analyses

involving the linguistic features of the stories. Then, we collected human ratings from 6 raters which independently judged all stories from all participants. We asked them to rate with a 5-point Likert scale (ranging from "not at all" as 1 to "very much" as 5) each story with their related triplet target words on three scales, namely ordinary, random, and creativity.

Next, we analyzed behavioral data by means of recent advances in the NLP field, using deep neural network models trained on large corpora of text data[53,54]. We opted for the adoption of these deep language models because of their unique ability to handle complex textual data such as stories[71,72]. Thus, we leveraged these models to extract metrics from the data that can be seen as an operationalization of the features we asked participants to control in the story generation, namely the novelty and appropriateness of the story. For the novelty feature, we used word2vec models[31,53] from the distributional semantics framework to extract word embedding vectors from the stories. There are mostly two versions of this model family, called Continuous Bag Of Words (CBOW) and Skip-gram. Both models employ a sliding window approach to move through text corpora. CBOW maximizes the probability of the target given the context, while Skip-gram is trained vice versa[100]. Notably, there is evidence in the literature that the hidden representations of these models highly correlate with judgments of human relatedness[73]. We used 3 pre-trained models trained using different methods (CBOW or Skip-gram) and different Italian text corpora for methodological robustness reasons. The first was a CBOW model trained using fastText on a concatenation of the Common Crawl and Wikipedia[105], while the second model was trained using the Skip-gram approach on Wikipedia[106]. The third model was also a Skip-gram model trained using fastText on the OpenSubtiles corpus[107]. All the models we used here were trained with a window size of 5, 10 negatives, and an embedding dimensionality of 300. We extracted word embeddings of the target and non-target words present in each story, including verbs, adjectives, and nouns in their lemmatized form. We exclude from this analysis stop words. Thus, by letting $\phi_t$ and $\phi_{nt}$ represent the sets of target and non-target words in the stories, the semantic distance metric $\mathscr{SD}$ is defined as the average cosine distance between each target word $w_i \in \phi_t$ and all non-target words $w_j \in \phi_{nt}$:

$$ SD = \frac{1}{|\phi_t|} \sum_{w_i \in \phi_t} \frac{1}{|\phi_{nt}|} \sum_{w_j \in \phi_{nt}} 1 - \frac{\sum_{k=1}^{N} w_i^k w_j^k}{\sqrt{\sum_{k=1}^{N} w_i^{k^2}} \sqrt{\sum_{k=1}^{N} w_j^{k^2}}} \tag{1} $$

where $|\phi_t|$ and $|\phi_{nt}|$ denote the cardinality of the target and non-target word sets, respectively, and $N$ represents the embedding dimensionality.

For the appropriateness feature, we capitalized on current state of the art transformer architectures for NLP[54,75], namely the Bidirectional Encoder Representations from Transformers (BERT) model. Trained on vast amounts of text data, BERT captures complex syntactic and semantic dependencies between words, accounting for the full context of a sentence by analyzing words bidirectionally (i.e., in both forward and backward directions). It does so by employing the Transformer[75], a deep neural network architecture that operates on the mechanisms of attention, enabling the model to focus on various elements of the text. For the same methodological robustness reasons as above, we used 2 instances of the model from HugginFace[108], namely the Italian cased BERT-XXL[109], trained on Wikipedia and the OPUS corpora[110], and BERTino[111], a distilBERT model[112] trained on the Paisa and ItWaC corpora[102]. These models allowed us to mask the target words from each story and let the model predict the most appropriate word to fill the masked tokens. Given a sequence of words as in our stories, the BERT model inputs the "tokenized" version of the story $S = \{\ell_1, \ell_2, ..., \ell_M\}$, where $i_i$ is the $i$-th token in the story and $M$ is the number of tokens. For the set of target words $\phi_t \subseteq S$, we computed the negative log-likelihood (NLL) of observing each target token $\ell_i \in \phi_t$ in its context as provided by the BERT output. Thus, for a given context $\mathscr{C}(\ell_i)$ around the target token $\ell_i$, where $\mathscr{C}(\ell_i) = S \setminus \{\ell_i\}$, BERT generates the probability distribution $P(\ell_i \mathscr{C}(\ell_i))$ over the possible words at the place of the masked $\ell_i$. Thus, we computed our surprise metric $\delta$ as the average NLL

across all target words in a story:

$$\delta = \frac{1}{|\phi_t|} \sum_{\ell_i \in \phi_t} -log\, P\left(\ell_i \mathscr{C}(\ell_i)\right) \tag{2}$$

## MRI data acquisition

Imaging data were acquired using a 4 T Bruker MedSpec Biospin MR scanner with a birdcage transmit and 8-channel receive head radio-frequency coil. The scanning duration of each fMRI session was approximately 12 min (730 volumes). fMRI images were acquired with a single shot T2*-weighted gradient-recalled echo-planar imaging (EPI) sequence (TR = 1000 ms, voxel resolution = $3 \times 3 \times 3$ mm$^3$, TE = 28 ms, FA = 59°, FOV = $210 \times 210$ mm$^2$; slice gap, 0 mm). Moreover, a structural T1-weighted anatomical scan was acquired (MP-RAGE; $1 \times 1 \times 1$ mm$^3$; FOV, $256 \times 256$ mm$^2$; 176 slices; GRAPPA acquisition with an acceleration factor of 2; TR, 2530 ms; TE1 = 1.64 ms, TE2 = 3.5 ms, TE3 = 5.36 ms, TE4 = 7.22 ms; inversion time (TI), 1100 ms; 7° flip angle).

## Anatomical data preprocessing

Results included in this manuscript come from preprocessing performed using *fMRIPrep* 23.1.3 (RRID:SCR_016216)[113,114], which is based on *Nipype* 1.8.6 (RRID:SCR_002502)[115]. A total of 1 T1-weighted (T1w) images were found within the input BIDS dataset. The T1-weighted (T1w) image was corrected for intensity non-uniformity (INU) with N4BiasFieldCorrection[116], distributed with ANTs (RRID:SCR_004757)[117], and used as T1w-reference throughout the workflow. The T1w-reference was then skull-stripped with a *Nipype* implementation of the *antsBrainExtraction.sh* workflow (from ANTs), using OASIS30ANTs as target template. Brain tissue segmentation of cerebrospinal fluid (CSF), white-matter (WM) and gray-matter (GM) was performed on the brain-extracted T1w using *fast* (FSL, RRID:SCR_002823)[118]. Volume-based spatial normalization to one standard space (MNI152NLin2009cAsym) was performed through nonlinear registration with antsRegistration (ANTs), using brain-extracted versions of both T1w reference and the T1w template. The following template was selected for spatial normalization and accessed with *TemplateFlow* (23.0.0)[119]: *ICBM 152 Nonlinear Asymmetrical template version 2009c* (RRID:SCR_008796; TemplateFlow ID: MNI152NLin2009cAsym)[120].

## Functional data preprocessing

For each of the 3 BOLD runs found per subject (across all tasks and sessions), the following preprocessing was performed. First, a reference volume and its skull-stripped version were generated using a custom methodology of *fMRIPrep*. Head-motion parameters with respect to the BOLD reference (transformation matrices, and six corresponding rotation and translation parameters) are estimated before any spatiotemporal filtering using *mcflirt* (FSL)[121]. BOLD runs were slice-time corrected to 0.452 s (0.5 of slice acquisition range 0s-0.905 s) using *3dTshift* from AFNI (RRID:SCR_005927)[122]. The BOLD time-series (including slice-timing correction when applied) were resampled onto their original, native space by applying the transforms to correct for head-motion. These resampled BOLD time-series will be referred to as *preprocessed BOLD in original space*, or just *preprocessed BOLD*. The BOLD reference was then co-registered to the T1w reference using *mri_coreg* (FreeSurfer) followed by *flirt* (FSL)[123] with the boundary-based registration[124] cost-function. Co-registration was configured with six degrees of freedom. Several confounding time-series were calculated based on the *preprocessed BOLD*: framewise displacement (FD), DVARS and three region-wise global signals. FD was computed using two formulations following Power (absolute sum of relative motions)[125] and Jenkinson (relative root mean square displacement between affines)[121]. FD and DVARS are calculated for each functional run, both using their implementations in *Nipype* (following the definitions by Power et al.[125]). The three global signals are extracted within the CSF, the WM, and the whole-brain masks. Additionally, a set of physiological regressors was extracted to allow for component-based noise correction (*CompCor*)[126]. Principal components are estimated after high-pass filtering the *preprocessed BOLD* time-series (using a discrete cosine filter with 128 s cut-off) for the two *CompCor* variants: temporal (tCompCor) and anatomical (aCompCor). tCompCor components are then calculated from the top 2% variable voxels within the brain mask. For aCompCor, three probabilistic masks (CSF, WM and combined CSF + WM) are generated in anatomical space. The implementation differs from that of Behzadi et al.[126] in that instead of eroding the masks by 2 pixels on BOLD space, a mask of pixels that likely contain a volume fraction of GM is subtracted from the aCompCor masks. This mask is obtained by thresholding the corresponding partial volume map at 0.05, and it ensures components are not extracted from voxels containing a minimal fraction of GM. Finally, these masks are resampled into BOLD space and binarized by thresholding at 0.99 (as in the original implementation). Components are also calculated separately within the WM and CSF masks. For each CompCor decomposition, the $k$ components with the largest singular values are retained, such that the retained components' time series are sufficient to explain 50 percent of variance across the nuisance mask (CSF, WM, combined, or temporal). The remaining components are dropped from consideration. The head-motion estimates calculated in the correction step were also placed within the corresponding confounds file. The confound time series derived from head motion estimates and global signals were expanded with the inclusion of temporal derivatives and quadratic terms for each[127]. Frames that exceeded a threshold of 0.5 mm FD or 1.5 standardized DVARS were annotated as motion outliers. Additional nuisance timeseries are calculated by means of principal components analysis of the signal found within a thin band (*crown*) of voxels around the edge of the brain, as proposed by Patriat et al.[128]. The BOLD time-series were resampled into standard space, generating a *preprocessed BOLD run in MNI152NLin2009cAsym space*. First, a reference volume and its skull-stripped version were generated using a custom methodology of *fMRIPrep*. All resamplings can be performed with *a single interpolation step* by composing all the pertinent transformations (i.e., head-motion transform matrices, susceptibility distortion correction when available, and co-registrations to anatomical and output spaces). Gridded (volumetric) resamplings were performed using antsApplyTransforms (ANTs), configured with Lanczos interpolation to minimize the smoothing effects of other kernels[129]. Non-gridded (surface) resamplings were performed using *mri_vol2surf* (FreeSurfer).

## fMRI data analysis

After generally preprocessing the signal with *fMRIPrep*, we proceeded to analyze the fMRI data using the following steps. First, we cleaned the images by performing linear detrending, low-pass (butterworth with a cutoff of 0.09 Hz), and high-pass (butterworth with a cutoff of 0.008 Hz) filtering and confounding regression analyses on the preprocessed BOLD signal, in this exact order. We included in the confounder variables taken from *fMRIPrep* the six rigid-body motion parameters (three translations and three rotations) and the estimated global, cerebrospinal fluid (CSF) and white matter signal, alongside the first derivatives of all these variables, resulting in a total of 18 variables. We performed all the analyses using the preprocessed BOLD signal, instead of using a conventional general linear model (GLM) approach. We made this choice since the period of interest (i.e., the ideation period) lasted for 30 seconds and we considered the adoption of a GLM approach unreasonable. This is because it would have implied that voxel activations should have lasted (on average) for such long time period. Thus, we extracted only the volumes belonging to the ideation period (30 in our case since we had a TR of 1 second), for each participant, condition, and trial. As a sanity check, we computed the variance of the signal across trials (stories) along each volume for each participant and condition to confirm that using the BOLD signal in the whole ideation period was capturing relevant brain activity. We also averaged the results across voxels in both the whole brain and all the 7 regions of interest (ROI) taken from the Yeo atlas[76].

Then, we computed the inter-subject correlation (ISC)[63,64] to investigate the similarity of the neural dynamics during the ideation period across

participants. To this end, we used the Schaefer atlas[130] with 400 ROIs and averaged the activity of the voxels within each ROI. Then, for each condition, we computed the correlation matrix between all pairs of participants in each ROI, using the entire BOLD time-series of the ideation period. Crucially, we computed the correlation matrix only using the stories across participant who were cued by the same target word triplets and average the results across target word triplets. Finally, we averaged the lower triangle of the correlation matrix to obtain a ISC value for each ROI.

Next, we performed Multivariate Pattern Analysis (MVPA) on the BOLD signal of the ideation period using the searchlight approach[77]. We used the Schaefer atlas[130] with 400 ROIs to select the voxels as local features on which performing the appropriate analysis. Thus, we conducted classification analyses using the support vector machine model with the radial basis function as kernel (rbf-SVM) and a regularization value C of 1. Model performance was evaluated using classification accuracy. We classified all the possible pairs of conditions at the group-level, by concatenating all the stories from all participants. In other words, for each selected ROI we had a matrix with as many rows as multiplying the number of participants, two times the number of stories (since was a binary classification) per condition and the number of volumes per story, and as many columns as the number of voxels in each ROI. We opted for performing group-level analysis because it allows us to identify population-wise invariants from neural activity patterns[78] and to control for the effects of the target words that were counterbalanced across participants. Similarly, we performed regression analyses to investigate where the semantic distance and surprise features were mostly encoded using the linear regression model with the ordinary least square estimator. Model performance was evaluated using mean square error.

Then, we continued our data analysis by applying the Connectome Harmonic Decomposition (CHD)[66] framework to our data. CHD decomposes a whole-brain activation signal in terms of harmonic modes of the human connectome. Each connectome harmonic mode quantifies to what extent the functional brain activity patterns deviate from the organization of the underlying structural connectome[66,68] based on its granularity (i.e., its spatial frequency value). Thus, CHD is a complementary approach to our previous searchlight analysis, since each harmonic is a whole-brain pattern with a characteristic spatial scale while in the searchlight approach, the implicit is that the brain activity can be seen as a composition of signals from localized units. We started our workflow by computing a template structural connectivity matrix from openly available data since we did not record individual DTI data. We used data from 88 healthy control individuals participating in the Early-Stage Schizophrenia Outcome study[131]. Individual structural connectivity matrices were already provided and arranged according to the Automated Anatomical Labeling (AAL) atlas[132] with 90 ROIs. We averaged them across participants, symmetrized the resulting matrix by summing it with its transposed and dividing by 2, and binarize it with a threshold corresponding to the average value across all connections. Then, we used surface reconstructed data from the *fsaverage*[133] as the template brain surface. We used the pial reconstructed surface with 5124 vertices for both hemispheres. With this data, we constructed the template structural connectome as a binary adjacency matrix $A$, using each cortical surface vertex as a node: for each pair $i$ and $j$ of the $n = 5124$ cortical nodes, each entry $A_{i,j}$ was set to 1 if there was a white matter tract connecting them and 0 otherwise. We achieved this by transposing the data from the previously computed structural matrix to the template adjacency matrix $A$ via identifying the regions in *fsaverage* of the AAL atlas and connecting them accordingly. This procedure accounted for the global or long-range structural connections given by the white matter tracts that wire the brain areas. But we also accounted for short-range grey-matter connections by setting to 1 entries of the matrix $A$ whose distance between the nodes was less than 10 mm, since it has been shown the importance of accounting for both local grey-matter and long-range white-matter structural connectivity patterns for the estimation of the connectome harmonics[134]. We projected the preprocessed BOLD signal to the cortical surface coordinates using the registration fusion method[135], which allows accurate projection from standard

volumetric coordinates to *fsaverage* cortical surface. Then, for extracting the connectome harmonics from the template structural connectome, we defined the degree matrix $D$ of the structural graph as:

$$D_{i,i} = \sum_{j=1}^{n} A_{i,j} \tag{3}$$

We then calculated the symmetric graph Laplacian $\Delta_G$ on the template adjacency matrix $A$ to estimate the Laplacian of the human structural connectome:

$$\Delta_G = D^{-\frac{1}{2}} L D^{-\frac{1}{2}}, \text{ with } L = D - A \tag{4}$$

Thus, we computed the connectome harmonics $\varphi_k$, $k \in \{1, ..., n\}$ by solving the following eigenvalue problem:

$$\Delta_G \varphi_k(v_i) = \lambda_k \varphi_k(v_i) \forall v_i \in V, \text{ with } 0 < \lambda_1 < \lambda_2 < \ldots < \lambda_n \tag{5}$$

where $\lambda_k$ is the corresponding eigenvalue of the eigenvector $\varphi_k$ and $V$ is the set of cortical vertices. Once obtained the connectome harmonics, we decomposed the spatial pattern of neural activity $F_t(v)$ over vertices $v$ and time (volume) $t$ as a linear combination of the set of connectome harmonics $\Psi = \{\varphi_k\}_{k=1}^{n}$:

$$F_t = \sum_{k=1}^{n} \omega_k(t) \varphi_k(v) \tag{6}$$

with the weight $\omega_k(t)$ of each connectome harmonic $\varphi_k$ at volume $t$ was estimated as the projection of the fMRI data $F_t(v)$ onto $\varphi_k$ via the dot product operation $\langle . \rangle$:

$$\omega_k(t) = \langle F_t, \varphi_k \rangle \tag{7}$$

Thus, our new data representation was a matrix for each story with rows as volumes and with columns as the weights $\omega_k(t)$ assigned to each connectome harmonic mode sorted in ascending order from the values of the corresponding eigenvalues $\lambda_k$. We conducted MVPA analyses such as classification and regression analyses with the same hyperparameters as in the searchlight analysis to investigate how spatial frequency patterns differed between each pair of conditions and how the behavioural features were encoded in the connectome harmonic space. For this, we adopted a sliding window approach by selecting for each run the 4% of the total harmonics (i.e., 205 harmonics) and a step size equal to the 2%.

In addition, we conducted functional connectivity analysis[69] to quantify the strength of functional coupling between specific brain areas during the story ideation period. Thus, we averaged the activity of the voxels within each of the 400 ROIs in the Schaefer atlas[130] and defined the functional connectivity measure as the inverse hyperbolic tangent function (arctanh) of the Pearson correlation coefficient between the two selected ROI time-series, for each story and participant. We average all the connectivity matrices computed per story within a participant to have one matrix per participant. Based on our results from the searchlight analysis and previous findings in the literature about semantic control, we selected three networks of areas using the Yeo atlas[76], namely the Default Mode Network (DMN), the Salience Attention Network (SAN) and the Frontoparietal Control Network (FCN). We computed the functional connectivity measure both within and between these networks for all possible pairs of ROIs and averaged the resulting values.

Finally, we also investigated the temporal irreversibility of the neural dynamics[98,136–138] in the ideation period using time-lagged directed connectivity analysis at the whole-brain across all brain areas. The key idea is to detect the temporal irreversibility (the arrow of time) through the asymmetry in the causal relationships between forward and artificially reversed backward time series. Specifically, we assessed temporal irreversibility between two ROI zero-meaned time series $ROI_i(t)$ and $ROI_j(t)$ and their

reversed backward version $ROI_i^{(r)}(t)$ and $ROI_j^{(r)}(t)$, obtained by flipping the time ordering. For forward and time-reversed evolution, the causal dependency is measured via the time-lagged correlation, for each story and participant, defined as:

$$C_f(\tau) = \text{corr}\left( ROI_i(t), ROI_j(t+\tau) \right) \tag{8}$$

$$C_r(\tau) = \text{corr}\left( ROI_i^{(r)}(t), ROI_j^{(r)}(t+\tau) \right) \tag{9}$$

Notably, if the system is reversible, then the lag $\tau$ will not induce any temporal asymmetry in the cross correlation between the two ROIs. We computed $C_f(\tau)$ and $C_r(\tau)$ across all pairs of the 400 ROIs (averaging the activity of all voxels in that region) in the Schaefer atlas[130], with the lag $\tau$ ranging from 1 to 15. We opted for this lag range since the ideation period consisted of 30 volumes, thus at the longest delay there are enough samples to compute the cross correlation in a reliable way. We then calculated the irreversibility matrix as the squared difference between the forward and reversed time-lagged correlations:

$$R(\tau) = \left( C_f(\tau) - C_r(\tau) \right)^{\circ 2} \tag{10}$$

where $^{\circ 2}$ stands for element-wise squaring operation. We average all the irreversibility matrices computed per story within a participant to have one matrix per participant. To examine the level of irreversibility between conditions across the lag dimension, we split the irreversibility values in short (from lag 1 to 7) and long (from lag 8 to 15) delay clusters and average the results.

## Statistics and Reproducibility
Statistical comparisons of the behavioral data ($N = 24$) were assessed by means of one-tail paired t-tests ($\alpha = 0.05$ if unspecified) as a statistical test and Cohen's d as a measure of effect size alongside its 95% confidence interval values. For the fMRI data analysis ($N = 24$), statistical analysis of the intersubject correlation was carried out using a permutation approach. A null distribution was generated by creating surrogate data via circularly shifting the time courses of the participants in the ideation period and repeating the ISC with 5000 iterations. P-values were defined as the proportion of permuted ISC values that were exceeded the observed ISC value at each ROI. Statistical analysis of the classification and regression MVPA analyses were carried out using the $5 \times 2$ cross-validation F-test[139]. We repeated 5 times a two-fold cross validation (i.e., 50% split) and measured the performance of the model against a model trained with permuted labels for 100 random permutations. Thus, we computed the pseudo f-statistic and the p-value using an F distribution with 10 and 5 degrees of freedom and used the difference between the models' performance as a measure of effect size. Statistical analysis of the functional connectivity results was carried out with the same approach as for the behavioral data. Multiple comparison correction was performed using False Discovery Rate (FDR) correction[140].

## Reporting summary
Further information on research design is available in the Nature Portfolio Reporting Summary linked to this article.

## Data availability
Minimally preprocessed behavioral and fMRI data for reproducing the results of this study are openly available at Open Science Framework (OSF)[141] with the following link https://osf.io/gxfpc/. Source data are provided as Supplementary Data.

## Code availability
The deep language models utilized in this study are available at the following links: - The first word2vec model was trained with the CBOW

on Italian Common Crawl and Wikipedia using fastText. https://fasttext.cc/docs/en/crawl-vectors.html - The second model was trained with the Skip-gram method on Italian Wikipedia. https://wikipedia2vec.github.io/wikipedia2vec/pretrained/ - The third model was trained with the Skip-gram method using fastText on the Italian OpenSubtiles corpus. https://github.com/jvparidon/subs2vec - The first BERT transformer model bert-base-italian-xxl-uncased is available on HugginFace at https://huggingface.co/dbmdz/bert-base-italian-xxl-uncased - The second DistilBERT model named BERTino is available on HugginFace at https://huggingface.co/indigo-ai/BERTino.

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

## Acknowledgements

We thank Elisa Guastella for her help during the data collection. This study was supported by a grant from the Italian Ministry of University and Research (Excellence Department Grant awarded to the Department of Psychology and Cognitive Science, University of Trento, Italy).

## Author contributions

C.R.: Conceptualization, Methodology, Investigation, Data curation, Software, Formal analysis, Visualization, Writing – original draft, Writing – Review & Editing. A.G.: Software, Formal analysis, Visualization, Writing – original draft, Writing – Review & Editing. CF: Conceptualization, Methodology, Supervision, Writing – Review & Editing. G.P.: Investigation. C.B.: Formal analysis, Supervision, Writing – Review & Editing. NDP: Conceptualization, Methodology, Supervision, Project administration, Funding acquisition, Writing – Review & Editing.

## Competing interests

The authors declare no competing interests.
