## [Transparent Peer Review file · Communications Biology]

Neural dynamics of semantic control underlying generative storytelling

Corresponding Author: Ms Clara Rastelli

Version 0:

Reviewer comments:

Reviewer #1

(Remarks to the Author)

This paper employs an ecological setting of generative storytelling while collecting participants' fMRI data to study the neural dynamics of semantic control. Although the paper presents the use of the ideation period as an innovation to avoid motion and verbal confounds, careful consideration must be given to its application. Several issues need to be addressed, such as ensuring that participants are actively generating stories during the ideation period, rather than merely thinking of a topic and generating sentences only during the reporting period. Additionally, it is worth investigating whether using the entire 30-second ideation period is more effective, or if focusing on the later stages of this period, which show lower variability according to the experimental results, would be more appropriate. The potential individual differences during this period also warrant examination. Finally, any findings related to the report period should be explored.

This paper employs NLP models, including Word2Vec and BERT, to generate quantitative scores for the novelty and appropriateness of the generated narratives. However, the model-generated scores differ from human evaluations, as no significant differences were observed between CR and RA, or between CR and OR. This raises doubts about the suitability of using these models to evaluate the novelty and appropriateness of narratives. Several questions need to be addressed: What is the individual correlation between participants? Are there specific participants responsible for the lack of significant results? What is the correlation between model-generated scores and human annotations? Additionally, for the BERT model, while it assesses the most appropriate word to fill masked tokens, a more natural approach to evaluating the appropriateness of narratives might be to calculate the perplexity of the narratives. This would allow for the use of more advanced models, such as the state-of-the-art GPT models.

Regarding the experimental results and discussions, the introduction of this paper references related works, including the distinct role of the semantic control network compared to brain areas encoding semantic representations and previous findings that did not implicate parietal areas. The implications of this study's findings for these questions should be explored in greater detail. Additionally, the conclusion, "Taken together, these findings demonstrated that the neural dynamics of the whole brain are organized in specific spatial frequency patterns during the ideation of complex linguistic outcomes, such as stories requiring different abstract features to be incorporated," could be made more specific to enhance clarity.

Furthermore, I am uncertain whether "our measure of appropriateness, derived from the BERT model's surprise, can be a valid measure for future studies attempting to quantify the usefulness of a narrative." There is existing literature in computational linguistics on calculating the appropriateness and usefulness of a narrative that may need to be addressed here.

Reviewer #2

(Remarks to the Author)

This article reports on an analysis of novel storytelling using NLP and neuroimaging. Participants are asked to tell Ordinary, Creative or Random stories. The analysis suggests the instructions are followed and that different areas of the brain are involved in various processes.

I found the work interesting, but felt it was largely exploratory and in need of more theory to frame the overall results. I think there are valuable findings to be made here, but I had trouble discerning them in the present form. My specific suggestions

are below.

1. It was difficult to discern what the overall patterns should be based on prior research. The work seems largely an exploratory analysis, but does it support any hypotheses? I wasn't sure exactly what the hypotheses were or from what literature they were drawn. More organization of the reader's expectations and clarity with respect to the hypotheses is needed. This work could help organize the field and really push the field forward, but in its current form it lacks focus.
2. One hypothesis that could be more clear is that executive control process support exploration and creative ideation, but should this also be true for random exploration? Is creative ideation more 'exploratory' than OR but less exploratory than random? I couldn't follow what the expectations were or what the results showed in light of the above question.
3. Do the measures in the brain correlate with the semantic distance measures in the NLP task? It appears you are correlating areas of the brain associated with those features, but do they also discriminate different story types? Should they?
4. I didn't understand the value of the spatial frequency analysis. What is this correlated with? What behavioral aspects does it indicate? Is CR between OR and RA? If not, this suggests perhaps a higher dimensionality, but can you indicate what those dimensions are? This last point is really valuable, as it may be a way to clearly identify what distinguishes creative from random. I expect a kind of inverted U-shaped function here, but given the results I expect this to be a trade-off between two different components. See Hills & Kenett 2024 below.
5. The overall narrative arc and theoretical underpinning of the work is left too undefined. There are established theories in the creativity literature that could be brought to bear here that would help ground the exploratory analysis better.
6. Ovando-Tellez has done some recent work on neuroimaging and creativity, which takes a somewhat similar approach across a number of tasks. I would imagine that this would offer some mapping for the neuroimaging results, and even some hypotheses to help frame the present work.
Ovando-Tellez, M., Benedek, M., Kenett, Y. N., Hills, T., Bouanane, S., Bernard, M., ... & Volle, E. (2022). An investigation of the cognitive and neural correlates of semantic memory search related to creative ability. *Communications Biology*, 5(1), 604.
7. Hills & Kenett 2024 recently presented a model on the interface between cognitive control and structure. This work makes explicit hypotheses about the role of cognitive control and semantic structure in creativity, and why CR might be different from OR and RA. Given the substantial work from Kenett and others arguing that creativity is structural, and from Benedek arguing that creativity is a control process, it's important to attempt to integrate these two ideas in the present work, and it appears there is room to achieve that.
Hills, T. T., & Kenett, Y. N. (2024). An entropy modulation theory of creative exploration. *Psychological Review*.

Version 1:

Reviewer comments:

Reviewer #1

(Remarks to the Author)

The responses address all of my concerns, and I don't have any further questions.

Reviewer #2

(Remarks to the Author)

Overall, I appreciate the revision. However, I think the section on entropy and irreversibility is incorrect and needs to be revised.

Starting on line 460: "At this point, entropy production becomes zero because the system no longer changes its distribution of states." This is not specific enough to be meaningful, or it is wrong. If a system has maximum entropy, then anything it produces is also maximum entropy (hardest to predict). High entropy systems produce high entropy data, by definition. The authors might be referring to the fact that a perfectly disordered system cannot be more disordered, but this is not as far as I can tell what they are measuring. It also doesn't seem particularly relevant. As noted by citation 104, higher entropy refers to more possible states, which means more difficulty in predicting past and future states.

461: "Equilibrium also satisfies detailed balance: every forward transition between states has a corresponding reverse transition." Not if it is in thermodynamic equilibrium--any output of the system is minimally predictive of what has come before. So reversible in what sense? However, I think the authors are referring to an aggregate measure here, where a high entropy system looks like itself in the previous timestep. However, the language here is not sufficiently clear to make this claim meaningful.

"Once the system leaves equilibrium, detailed balance breaks down, and the system's dynamics become irreversible. We highlighted this distinction since there is reported in the literature evidence for increased resting-state entropic brain dynamics in divergent thinking (104)." Yes, and this is also predicted by Hills & Kenett (36), who demonstrate how this a mechanism for executive control's contribution to explaining a common finding across the creativity literature.

"Our irreversibility results contrast these previous findings, since higher irreversibility are linked to lower entropic states." This is unclear and seems wrong. I do not think the present results are inconsistent with 104 or 36. As I understand the results, the CREative condition is more irreversible. In classical thermodynamics, statistical mechanics, and information

entropy, higher irreversibility is associated with higher entropy. Previous states do not predict past states in high entropy systems. Therefore, the CReative condition with higher irreversability appears to be higher entropy, consistent with 104 and 36.

To me, this is the most important result in the article along with what brain regions might be involved in this entropy modulation. I don't see an inconsistency with past literature. Am I missing something?

Version 2:

Reviewer comments:

Reviewer #2

(Remarks to the Author)

Thanks to the authors for their response. I appreciate them clearing things up for me. The article is far above my threshold for publication. I think the issue of reversibility and entropy will remain a bit confusing for some readers, but things are clear enough from context and I now get that the authors are referring specifically to entropy production. Higher irreversibility means the system is dynamically structured and directional, not necessarily more predictable or ordered. The authors make their point fairly clearly in the article and in their response, and placing this in relation to structured and hierarchical neural dynamics makes the point still clearer. The reference to Shinozuka also helped me quite a bit, as I now better understand the sense in which they mean irreversible. This remains a fascinating result, and agree that it seems inconsistent with 104 and Shinozuka et al, but does not seem to apply to 36. 36 uses Shannon entropy to explain cognitive flexibility, while the uploaded article uses entropy production to measure neural irreversibility. That was my confusion.

Response Letter

We thank the reviewers and the editor for their constructive feedback and for highlighting the potential of our work. Following the reviewers' thoughtful comments and suggestions, we have carefully revised the manuscript and conducted additional analyses where appropriate. We are confident that these revisions address all concerns raised by the reviewers and substantially enhance the quality and clarity of the manuscript.

Main changes:

- We added two additional critical analyses to strengthen the main claims of the paper. The first is about intersubject correlation, which elucidated the important cortical hubs emerging during story ideation. The second one is about the temporal irreversibility of the neural dynamics, highlighting the enhanced hierarchical structure of causal interactions as a neural signature of creative generative storytelling.
- We thoroughly revised the introduction to better articulate our hypotheses and the study's rationale, explicitly addressing the contextualization of our work as suggested by reviewers.
- We expanded the discussion to further clarify the implications of our findings, ensuring that all concerns raised by the reviewers were carefully addressed.
- We added several methodological details in the method section to improve replicability and extensively described the new analyses pipelines.

We appreciate the reviewers' and editor's guidance, which has helped us extensively improve the manuscript. We believe these changes address all the concerns raised and provide a more rigorous and contextually nuanced presentation of our findings.

To help reading our replies, the reviewers' comments are in **black** and our replies in **blue**.

Reviewers' comments:

Reviewer #1 (Remarks to the Author):

This paper employs an ecological setting of generative storytelling while collecting participants' fMRI data to study the neural dynamics of semantic control. Although the paper presents the use of the ideation period as an innovation to avoid motion and verbal confounds, careful consideration must be given to its application. Several issues need to be addressed, such as ensuring that participants are actively generating stories during the ideation period, rather than merely thinking of a topic and generating sentences only the reporting period.

We thank the reviewer for the insightful comments. Critically, our decoding analyses, performed on local brain activity and at different spatial frequencies to predict linguistic features extracted from the generated stories, provide compelling evidence that participants actively generated content during the ideation phase as instructed (Fig. 3d, Fig. 4d). If participants only reported the stories without ideating them during the ideation period, we could not have observed a decoding performance significantly higher than chance level. To further clarify task adherence, participants were explicitly instructed to report verbatim what they had thought during the ideation period, immediately following its conclusion. This instruction ensured that the verbalized stories accurately reflected the content generated during the ideation phase. We have now added the detailed version of the task instructions in the revised manuscript (see Methods, Procedure lines 526-538).

Additionally, it is worth investigating whether using the entire 30-second ideation period more effective, or if focusing on the later stages of this period, which show lower variability according to the experimental results, would be more appropriate.

We appreciate the reviewer's suggestion regarding our use of the full ideation period. Our choice was informed by several considerations. First, as noted, we examined trial-by-trial variability at each time point (volume) to assess whether the signal captured meaningful activity. A reduction in variability typically indicates that

participants converge on similar processing, which in our case presumably indicates returning to a resting or preparatory state before providing their narrative. This pattern emerged only at the very end of the ideation period, leading us to analyze the full duration, as participants utilized most of it. Additionally, potential delays in the hemodynamic response could influence the timing and amplitude of neural activity measured by the BOLD signal, making the selection of a specific a priori window imprudent. Finally, choosing one or multiple time windows would introduce additional hyperparameters and complexity to the analysis pipeline, without a clear benefit to justify the trade-off. We now added these considerations to the main text for further clarification to the reader (lines 203-205).

The potential individual differences during this period also warrant examination.

We thank the reviewer for this valuable suggestion. In response, we decided to perform an intersubject correlation (ISC) analysis to explore individual variability in neural dynamics during the ideation period. These results helped to shape a clearer picture of the neural dynamics underlying generative storytelling, by adding complementary evidence to the existing ones. We included the results of this analysis in the revised manuscript. Specifically, here we just report a short excerpt of the results section (lines 205-220): *“Therefore, we started analyzing the neural data by assessing the similarity of the neural dynamics underlying the generation of the stories by means of inter-subject correlation (ISC) analysis (130, 131). In essence, we asked which brain areas had a similar temporal profile in the ideation period across participants. Crucially, we computed ISC for each condition by separating the stories having the same triplet of target words and averaging the results across target word triplets. This helped us to directly quantify the degree of similarity across participants when ideating a story starting from the same linguistic constrains. [...] In sum, our findings highlight that during narrative generation, left frontal and visual cortices consistently showed inter-subject similarity across all conditions, while random story ideation elicited even higher alignment in parietal–frontal areas, underscoring distinct patterns of neural synchronization in response to different storytelling constraints.”*

Finally, any findings related to the report period should be explored.

While we acknowledge the reviewer’s suggestion to explore findings related to the report period, we intentionally designed our study to decouple story generation from articulation. This methodological choice aimed to eliminate movement artifacts associated with overt verbal expression and address limitations highlighted in prior research on generative storytelling (e.g., Liu et al., 2015, Human Brain Mapping). Incorporating the report period would reintroduce these artifacts, potentially confounding the neural signals of interest and detracting from the clarity of our findings.

This paper employs NLP models, including Word2Vec and BERT, to generate scores for the and appropriateness of the generated narratives. However, the model-generated scores differ from human evaluations, as no significant differences were observed between CR and RA, or between CR and OR.

This raises doubts about the suitability of using these models to evaluate the novelty and appropriateness of narratives. Several questions need to be addressed: What is the individual correlation between participants? Are there specific participants responsible for the lack of significant results? What is the correlation between model-generated scores and human annotations?

We thank the reviewer for raising this point. We would like to clarify that significant differences do, in fact, emerge from our analyses. Specifically, the semantic distance metric appears to effectively discriminate the element of novelty between CR and OR, as well as between RA and OR (Fig. 2b top), while the surprise metric successfully captures significant differences in appropriateness between CR and RA and between OR and RA (Fig. 2b bottom). The same patterns is present in the human raters (Fig. S1a), as the scales appropriately distinguished the expected features (e.g. the ordinary scale rated the ordinary condition significantly higher than creative and random). Importantly, we want to clarify that we added the human raters just as a confirmatory analysis more than a comparative one, since the sample size (N=6) of the raters was too low to be used for quantitative analyses. Also, we believe that the use of computational methods such as deep language models has the fundamental advantage of being reproducible, while relying on few human evaluators is usually an unstable measure.

Additionally, for the BERT model, it assesses the most appropriate word to fill masked tokens, a more natural approach to evaluating the appropriateness of might be to calculate the perplexity of the narratives. This would allow for the use of more advanced models, such as the state-of-the-art GPT models.

We thank the reviewer for the valuable suggestion. We would like to clarify that our methodological decision of using the BERT model was driven by the necessity to focus on the target words when analyzing the narratives. This choice stemmed from the constraints of our story generation task, where target words were required to appear in the participants' stories. By examining how participants integrated these target words, we were able to derive a more nuanced and task-specific measure of appropriateness, rather than a generic one.

While models such as GPT, trained on self-supervised frameworks like next-word prediction, are powerful, they were less suitable for our objectives. Their capacity to compute surprise is inherently limited to past tokens, which would not align with our need to assess target words in the context of the entire story. In contrast, BERT's masked language modeling framework allowed us to selectively mask the target words and compute the surprise based on the full narrative context, making it the optimal choice for our study.

Regarding the experimental results and discussions, the introduction of this paper references related works, including the distinct role of the semantic control network compared to brain areas encoding semantic representations and previous findings that did not implicate parietal areas. The implications of this study's findings for these questions should be explored in greater detail.

We are grateful to the reviewer for raising this important point. In response, we have updated the manuscript to provide a more detailed exploration of the implications of our findings, which now reads (lines 394-399): *"Interestingly, our observation of parietal involvement diverges from a recent meta-analysis (22) that found no parietal activation for semantic control. One likely explanation is that the meta-analysis primarily examined simpler, word-level or passive comprehension tasks, whereas our study employed an active, more complex narrative-generation paradigm. We speculate that this increased complexity may specifically recruit parietal regions that do not come into play when the task is limited to simpler or more passive forms of language processing"*

Additionally, the conclusion, "Taken together, these findings demonstrated that the neural of the whole brain are in specific spatial frequency patterns during the ideation of complex linguistic outcomes, such as stories requiring different abstract features to be incorporated," could be made more specific to enhance clarity.

We thank the reviewer for this valuable feedback, we now edited the sentence to improve clarity, which now reads: *"Taken together, these findings demonstrated that the neural dynamics of the whole-brain is organized in specific spatial frequency patterns, representing the decoupling of functional neural activity from the underlying structural connectivity, during the ideation of complex linguistic outcomes such as stories requiring different abstract features to be incorporated."*

Furthermore, I am uncertain whether "our measure of derived from the BERT model's surprise, can be a valid measure for future studies attempting to quantify the usefulness of a narrative." There is existing literature in computational on calculating the appropriateness and usefulness of a narrative that may need to be addressed here.

We thank the reviewer for the valuable comment. Indeed, we were a bit too unspecific in this passage of the text. We now revised the sentence to specify that this measure can be valuable only in the field of creative cognition and not in general, which reads: *"our measure of appropriateness, derived from the BERT model's surprise, has not been used yet and in our opinion can be a valid measure for further studies attempting at quantifying the usefulness of a narrative in the creative cognition field."*

Reviewer #2 (Remarks to the Author):

This article reports on an analysis of novel storytelling using NLP and neuroimaging. Participants are asked to tell ORdinary, CReative or RANdom stories. The analysis suggests the instructions are followed and that different areas of the brain are involved in various processes.

I found the work interesting, but felt it was largely exploratory and in need of more theory to frame the overall results. I think there are valuable findings to be made here, but I had trouble discerning them in the present form. My specific suggestions are below.

It was difficult to discern what the overall patterns should be based on prior research. The work seems largely an exploratory analysis, but does it support any hypotheses? I wasn't sure exactly what the hypotheses were or from what literature they were drawn. More organization of the expectations and clarity with respect to the

hypotheses is needed. This work could help organize the field and really push the field forward, but in its current form it lacks focus.

We thank the reviewer for their valuable feedback and for recognizing the potential of our work. We acknowledge the exploratory nature of the study, as generative storytelling has not been extensively investigated using a combination of NLP and neuroimaging methods. Given this novelty, we approached the study as a foundational step in this emerging area. Nonetheless, we now have made a concerted effort to situate our findings within the existing literature on semantic processing and neuroscience of creativity, which are reflected both in the revised introduction and discussion of the results. Crucially, we now extended the analysis pipeline by adding two important steps regarding the intersubject correlation and the temporal irreversibility of the neural dynamics underlying the narrative ideation. While the first one elucidated the important cortical hubs emerging during story ideation, the second one highlighted the enhanced hierarchical structure of causal interactions as a neural signature of creative generative storytelling.

One hypothesis that could be more clear that executive control process support exploration and creative ideation, but should this also be true for random exploration? Is creative ideation more 'exploratory' than OR but less exploratory than random? I couldn't follow what the expectations were or what the results showed in light of the above question.

We thank the reviewer for raising this important point. We have extensively now revised the introduction to clarify the hypotheses and rationale of the study. Specifically, regarding the expectations about the exploratory behavior in the conditions, we now added (lines 125-129): "*We hypothesized that this experimental manipulation influenced the exploration strategies of participants when navigating the semantic space to generate a narrative. Specifically, we hypothesized that the exploratory behavior should be minimal in the ordinary condition, lacking novelty, and maximal in the random condition, lacking appropriateness, with the creative condition having an intermediate level of exploration*".

Do the measures in the brain correlate with the semantic distance measures in the NLP task? It appears you are correlating areas of the brain associated with those features, but do they also discriminate different story types? Should they?

We thank the reviewer for raising this point. As shown in figure 3d, we found that the neural activations in specific regions of the left frontal and parietal cortex could significantly predict the semantic distance extracted from the stories. Also, as shown in figure 2b top, this semantic distance measure significantly differed among the conditions (RA > CR > OR). We believe this should be the case since this metric encodes the novelty component of the creative generation process, which given the instruction set of the task, should differ among the conditions. Please also note that, as shown in figure S1d, the semantic distance measure has a low correlation with the surprise measure, supposed to encode appropriateness, suggesting that these two measures really capture different dimensions of the stories.

I didn't understand the value of the spatial frequency analysis. What is this correlated with? What behavioral aspects does it indicate? Is CR between OR and RA? If not, suggests perhaps a higher dimensionality, but can you indicate what those dimensions are?

We thank the reviewer for raising this point. We adopted this Connectome Harmonic Decomposition (CHD, Fig. 4a) because, after having identified specific brain regions exhibiting significant activity through the searchlight analysis, we wanted to extend our investigation to a more system-level, whole-brain approach. As we also highlighted in the discussion section (lines 437-440): "*Note that, the transition from low to high spatial frequencies can be interpreted as a growing decoupling of functional neural activity from the underlying structural connectivity (94, 95). Therefore, the CHD framework allowed us to reveal the multi-scale network organization of the connectome as well as the asymmetric relationship between brain structure and function when participants performed naturalistic story generation*". To better clarify the results of the Connectome Harmonic Decomposition (CHD) analysis, we now extended the introduction section, which now reads (lines 138-142): "*Moreover, we also decomposed the neural dynamics underlying the story generation into its connectome harmonics and investigated the same research questions, using the recently proposed framework of Connectome Harmonics Decomposition (CHD) (70–72), and specifically asking whether there are distinct spatial frequencies that discriminate the different ideation modalities or encode the linguistic features*".

This last point is really valuable, as it may be a way to clearly identify what distinguishes creative from random. I expect a kind of inverted U-shaped function here, but given the results I expect this to be a trade-off between two different components. See Hills & Kenett 2024 below.

The overall narrative arc and theoretical underpinning of the work is leR too undefined. There are established theories in the creativity literature that could be brought to bear here that would help ground the exploratory analysis better. Ovando-Tellez has done some recent work on neuroimaging and creativity, which takes a somewhat similar approach across a number of tasks. I would imagine that this would offer some mapping for the neuroimaging results, and even some hypotheses to help frame the present work.

NI., Benedek, N., Kenett, Y. N., Hills, T., Bouanane, S., Bernard, M., ... & Volle, E. (2022). An investigation of the cognitive and neural correlates of semantic memory search related to creative ability. *Communications Biology*, 5(1), 604.

We thank the reviewer for raising this important point. We agree that the theoretical rationale was somehow missing in the previous submission. Now we extensively edit the introduction to improve this point, by specifically arguing the reasons why we designed this study and run these analyses. We also thank the reviewer for the suggested study, which we believe is an important work in the field connectome predictive modeling with creative abilities, and added it to our reference list.

Hills & Kenett 2024 recently presented a model on the interface between cognitive control and structure. This work makes explicit hypotheses about the role of cognitive control and semantic structure in creativity, and why CR might be different from OR and RA. Given the substantial work from Kenett and others arguing that creativity is structural, and from Benedek arguing that creativity is a control process, it's important to attempt to integrate these two ideas in the present work, and it appears there is room to achieve that.

Hills, T. T., & Kenett, Y. N. (2024). An entropy modulation theory of creative exploration. *Psychological Review*. We thank the reviewer for the valuable suggestion. We now extensively discussed the implication of our results for the role of semantic control in creative cognition. At the same time, though, we prefer to keep the paper open to a broader audience than solely the creative cognition community, since most of our results are also highly relevant for the linguistic and narrative community. We will surely incorporate the theoretical insights from the suggested reference in future investigations specifically tailored for the creative cognition community.

Response Letter

We thank the reviewers and the editor for their constructive feedback. Following the reviewers' thoughtful comments and suggestions, we have carefully revised the manuscript and are confident that these revisions address all concerns raised by the reviewers and enhanced the quality of the manuscript, especially the discussion section.

To help reading our replies, the reviewers' comments are in **black** and our replies in **blue**.

Reviewers' comments:

Reviewer #1 (Remarks to the Author):

The responses address all of my concerns, and I don't have any further questions.

We thank the reviewer for the valuable revision that substantially improved the paper

Reviewer #2 (Remarks to the Author):

Overall, I appreciate the revision. However, I think the section on entropy and irreversibility is incorrect and needs to be revised.

Starting on line 460: "At this point, entropy production becomes zero because the system no longer changes its distribution of states." This is not specific enough to be meaningful, or it is wrong. If a system has maximum entropy, then anything it produces is also maximum entropy (hardest to predict). High entropy systems produce high entropy data, by definition. The authors might be referring to the fact that a perfectly disordered system cannot be more disordered, but this is not as far as I can tell what they are measuring. It also doesn't seem particularly relevant. As noted by citation 104, higher entropy refers to more possible states, which means more difficulty in predicting past and future states.

We thank the reviewer for the appreciation of our revision. Maybe we were not entirely clear about what we meant by the word "production". By entropy production, we (and the statistical physics and thermodynamics field) refer to an increase in the entropy of the system. The reviewer commented that "If a system has maximum entropy, then anything it produces is also maximum entropy", but in the text we meant specifically the "entropy production" and not something "produced" by the system. In any case, we followed reviewer's suggestion to increase readability and now emphasize even more this statement by adding (line 461): "*In order words, if the entropy of the system is maximum, there is no chance to increase it*".

Moreover, as we outlined in that paragraph of the discussion, it is true that we did not measure whether the system reached the maximum entropy state, but we used this example to explain the difference between the entropy and entropy production concepts. We believe this is a crucial and relevant point to discuss since it is important to distinguish the two concepts of entropy and entropy production, as also evidenced by the reviewer's comments themselves to elucidate more on this issue. About the cited definition of entropy by ref 104, this is exactly how we defined entropy and in opposition to the measure of irreversibility we used, which indicate a more structured dynamics, thus less disordered, as we also pointed out in line 446-448: "*with higher irreversibility signaling a more pronounced hierarchy (more directed interactions) of causal interactions in the neural dynamics*".

461: "Equilibrium also satisfies detailed balance: every forward transition between states has a corresponding reverse transition." Not if it is in thermodynamic equilibrium---any output of the system is minimally predictive of what has come before. So reversible in what sense? However, I think the authors are referring to an aggregate measure here, where a high entropy system looks like itself in the previous timestep. However, the language here is not sufficiently clear to make this claim meaningful.

We thank the reviewer for raising this important point. We agree that the text may be misleading and corrected accordingly, which now reads (lines 462-466): "*At thermodynamic equilibrium, the probability distribution over the system's microstates becomes stationary, even though individual state transitions continue, the overall distribution remains unchanged, resulting in zero net entropy production. Once the system is driven away from*

equilibrium, the absence of detailed balance leads to entropy production and marks the emergence of temporal irreversibility." We now emphasize the fact that at equilibrium the distribution over system's possible states becomes stationary, thus being more predictable.

"Once the system leaves equilibrium, detailed balance breaks down, and the system's dynamics become irreversible. We highlighted this distinction since there is reported in the literature evidence for increased resting-state entropic brain dynamics in divergent thinking (104)." Yes, and this is also predicted by Hills & Kenett (36), who demonstrate how this a mechanism for executive control's contribution to explaining a common finding across the creativity literature. "Our irreversibility results contrast these previous findings, since higher irreversibility are linked to lower entropic states." This is unclear and seems wrong. I do not think the present results are inconsistent with 104 or 36. As I understand the results, the CReative condition is more irreversible. In classical thermodynamics, statistical mechanics, and information entropy, higher irreversibility is associated with higher entropy. Previous states do not predict past states in high entropy systems. Therefore, the CReative condition with higher irreversibility appears to be higher entropy, consistent with 104 and 36. To me, this is the most important result in the article along with what brain regions might be involved in this entropy modulation. I don't see an inconsistency with past literature. Am I missing something?

We thank the reviewer for raising this point. We believe that the critical issue here is that the reviewer claims that *"In classical thermodynamics, statistical mechanics, and information entropy, higher irreversibility is associated with higher entropy"*, which to our knowledge is an incorrect statement. As we discussed in the main text and in the above replies, higher irreversibility is *not* associated to higher entropy. Actually, it is usually the opposite, that higher irreversibility signals lower entropic state. Notably, this irreversibility measure has to be interpreted as a measure of structured and hierarchical neural dynamics, with higher values meaning higher structure and thus less entropy. To provide a critical example in the literature, Shinozuka et al. (2025, *Imaging Neuroscience*) studied the irreversibility of the neural dynamics during LSD psychedelic state with our same methodology. They found that LSD makes the neural dynamics significantly less irreversible than placebo (see for example their Fig. 2A), which is in lines with extensively validated findings in the psychedelics literature that LSD *increase* the entropy of the neural dynamics. Crucially, they even test this inverse relationship on the same dataset (Mediano et al., 2024 *ACS Chemical Neuroscience*). Thus, as also empirically showed, higher entropy is linked to lower irreversibility and vice versa. Crucially, we also elaborated that the result on the creative condition resulting more irreversible is actually very intuitive. As we stated (lines 451-454): *"This could be explained by the observation that creative stories required generally more effort than the control ones, since the need to balance two essential features as novelty and appropriateness, which is reflected in the higher need to orchestrate and structure the neural dynamics to reach the creative solution"*.